# Time-resolved phosphoproteomics reveals scaffolding and catalysis-responsive patterns of SHP2-dependent signaling

Vidyasiri Vemulapalli[1,2], Lily A Chylek[3], Alison Erickson[4], Anamarija Pfeiffer[5], Khal-Hentz Gabriel[1,2], Jonathan LaRochelle[1,2], Kartik Subramanian[3], Ruili Cao[1], Kimberley Stegmaier[6], Morvarid Mohseni[7], Matthew J LaMarche[7], Michael G Acker[7], Peter K Sorger[3], Steven P Gygi[4], Stephen C Blacklow[1,2]*

[1]Department of Cancer Biology, Dana-Farber Cancer Institute Boston, Boston, United States; [2]Department of Biological Chemistry & Molecular Pharmacology, Blavatnik Institute, Harvard Medical School, Boston, United States; [3]Laboratory of Systems Pharmacology, Harvard Medical School, Boston, United States; [4]Department of Cell Biology, Harvard Medical School, Boston, United States; [5]Novo Nordisk Foundation Center for Protein Research, University of Copenhagen, Copenhagen, Denmark; [6]Department of Pediatric Oncology, Dana Farber Cancer Institute, Boston, United States; [7]Novartis Institutes for Biomedical Research, Cambridge, United States

**Abstract** SHP2 is a protein tyrosine phosphatase that normally potentiates intracellular signaling by growth factors, antigen receptors, and some cytokines, yet is frequently mutated in human cancer. Here, we examine the role of SHP2 in the responses of breast cancer cells to EGF by monitoring phosphoproteome dynamics when SHP2 is allosterically inhibited by SHP099. The dynamics of phosphotyrosine abundance at more than 400 tyrosine residues reveal six distinct response signatures following SHP099 treatment and washout. Remarkably, in addition to newly identified substrate sites on proteins such as occludin, ARHGAP35, and PLCγ2, another class of sites shows reduced phosphotyrosine abundance upon SHP2 inhibition. Sites of decreased phospho-abundance are enriched on proteins with two nearby phosphotyrosine residues, which can be directly protected from dephosphorylation by the paired SH2 domains of SHP2 itself. These findings highlight the distinct roles of the scaffolding and catalytic activities of SHP2 in effecting a transmembrane signaling response.

*For correspondence:
stephen_blacklow@hms.harvard.edu

## Introduction

SHP2 is a non-receptor tyrosine phosphatase that is essential for mammalian development (*Saxton et al., 1997*). In humans, germline mutations of *PTPN11* cause the developmental disorders Noonan and LEOPARD syndromes (*Tartaglia and Gelb, 2005*). Somatic *PTPN11* activating mutations are also found frequently in juvenile myelomonocytic leukemia and, to a lesser extent, in solid tumors (*Bentires-Alj et al., 2004*).

Numerous studies have shown that SHP2 acts as a positive effector of receptor tyrosine kinase (RTK) signaling (*Bennett et al., 1994*; *Easton et al., 2006*; *Tang et al., 1995*). SHP2 facilitates the full induction of Ras-dependent extracellular signal-regulated kinase (ERK) proteins following stimulation of cells with epidermal growth factor (EGF) or other receptor tyrosine kinase ligands. SHP2 also serves as a positive regulator of numerous other signaling systems, including cytokine (*Xu and*

*Qu, 2008*), programmed cell death (*Yokosuka et al., 2012*), and immune checkpoint pathways (*Gavrieli et al., 2003*).

The SHP2 protein consists of two SH2 domains (N-SH2, C-SH2), followed by a phosphatase (PTP) domain, with a natively disordered C-terminal tail that contains tyrosine residues known to become phosphorylated (*Bennett et al., 1994*; *Keegan and Cooper, 1996*). X-ray structures of the SHP2 protein core, encompassing the two SH2 domains and the PTP domain, show that the wild-type protein normally adopts an autoinhibited conformation in which the N-SH2 domain occludes the active site of the PTP domain (*Hof et al., 1998*). Activation of the enzyme requires disengagement of the N-SH2 domain from the PTP domain and subsequent recruitment of SHP2 to phosphotyrosine docking sites on substrate proteins. Recruitment can rely on the N-SH2 domain, C-SH2 domain, or both domains. Most oncogenic mutations of SHP2 lie at the interface between the N-SH2 and PTP domains, thereby increasing the propensity of the enzyme to adopt an open, active conformation (*LaRochelle et al., 2018*; *LaRochelle et al., 2016*).

Numerous studies in a range of cell types have identified phosphotyrosine (pY) residues on adaptor proteins (e.g. GAB1 and GAB2 [*Arnaud et al., 2004*; *Cunnick et al., 2001*]) or on RTKs themselves (e.g. PDGFβR [*Rönnstrand et al., 1999*]) thought to recruit SHP2 to sites of active signaling following RTK activation (such sites include multiprotein complexes formed on RTK intracellular tails and IRS-1 adapters). Genetic and mutational studies have implicated specific pY residues on RTKs as putative SHP2 substrates (*Agazie and Hayman, 2003*; *Bunda et al., 2015*; *Klinghoffer and Kazlauskas, 1995*).

The availability of a potent and highly selective allosteric inhibitor of SHP2 now allows us to capture the whole proteome dynamics of the SHP2 dependence of the EGF response. Thus, we use here the allosteric inhibitor SHP099 (*Garcia Fortanet et al., 2016*) to study the role of SHP2 on phosphoprotein dynamics at the whole proteome level following EGF stimulation of a breast cancer cell line carrying an EGFR amplification. Global analysis of time-resolved changes in pY abundance at over 400 tyrosine residues reveals several distinguishable response signatures following SHP099 exposure and washout. Putative substrate sites fall into two classes, with increased pY abundance at early and/or late timepoints, respectively. Proteins in this category include previously unidentified substrate sites on proteins such as occludin, ARHGAP35, and PLCγ. Yet another class, which contains the largest number of dynamic pY sites, exhibits decreased pY abundance. These latter sites are enriched on proteins with two nearby pY residues, some of which are directly protected from dephosphorylation by the SH2 domains of SHP2. These data emphasize the two distinct and interrelated biochemical activities of SHP2 – dephosphorylation and phosphosite protection – and identify specific sites relevant to the activity of SHP099 and similar molecules such as TNO155 as therapeutic lead compounds (*LaMarche et al., 2020*).

## Results

### Dynamic regulation of the EGF-responsive phosphoproteome by SHP2

We investigated the role of SHP2 in the responsiveness to EGF by using the EGFR-amplified cell line MDA-MB-468, derived from a patient with triple-negative breast cancer. To identify timepoints for in-depth proteomic analysis, we monitored the influence of SHP2 on the response to EGF stimulation using ERK1/2 phosphorylation as a readout. Cell extracts were prepared at a series of timepoints from three treatment conditions: (1) following pretreatment with dimethyl sulfoxide (DMSO) and then stimulation with 10 nM EGF, (2) following pretreatment with the SHP2 inhibitor SHP099 for 2 hr and then stimulation with EGF, and (3) following pretreatment with SHP099 and then stimulation with EGF for 10 min, after which drug was washed out and medium containing EGF replenished. Because the response of the EGF receptor to EGF stimulation occurs in the 1–2 min time period (*Jadwin et al., 2016*), we focused attention on the 5–30 min time window to ensure the observation of SHP2-dependent events downstream of EGF receptor stimulation. Immunoblot analysis revealed the expected effect of EGF stimulation: a dramatic increase in phospho-ERK1/2 (p-ERK1/2) levels followed by a decline toward basal levels by 30 min. The induction of p-ERK1/2 was greatly attenuated by pretreatment of cells with 10 μM SHP099. SHP099 washout in the continued presence of EGF (condition 3) revealed p-ERK1/2 induction with a kinetic profile similar to that of EGF stimulation in

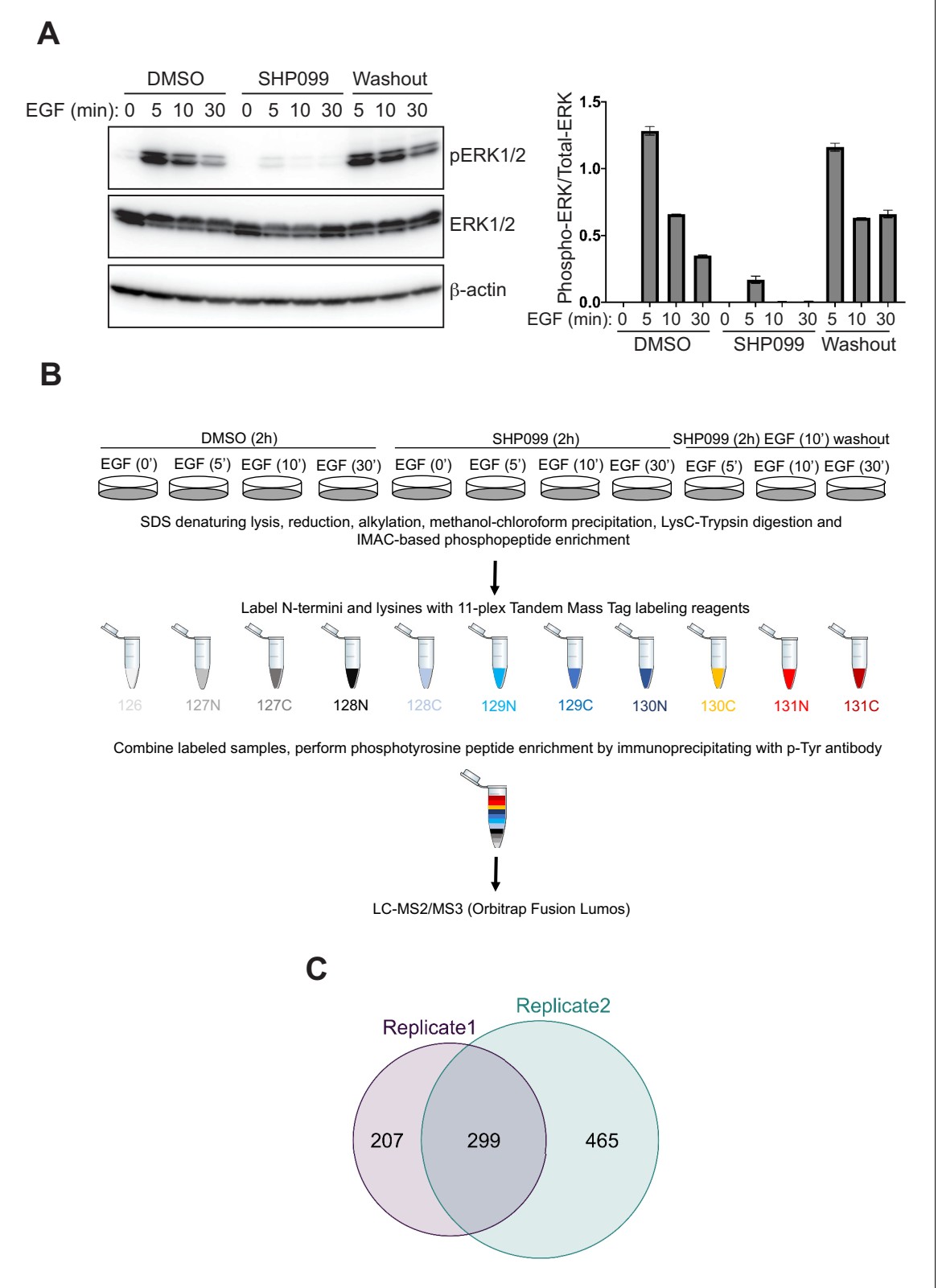

**Figure 1.** Phosphoproteomic studies in MDA-MB-468 cells. (**A**) Left: Western blot showing the phosphoERK1/2 abundance after EGF stimulation alone, EGF stimulation in the presence of SHP099, and EGF stimulation in the presence of SHP099, followed by drug washout 10 min after EGF stimulation. Right: quantification of the Western blot data, calculated from the ratio of [pERK1/2]/[ERK1/2] band intensities. (**B**) Schematic illustration of treatment

*Figure 1 continued on next page*

*Figure 1 continued*

conditions and mass spectrometry workflow. Western blot and phosphoproteomic data are both representative of two independent biological replicates (n = 2). (**C**) Venn diagram showing the overlap of phosphopeptides identified in biological replicates 1 and 2.

The online version of this article includes the following source data and figure supplement(s) for figure 1:

**Source data 1.** Table showing Phospho-ERK1/2 levels normalized to total-ERK1/2 levels (as quantified by measuring band intensities using ImageJ) in MDA-MB-468 cells treated with SHP099 and EGF as indicated.

**Figure supplement 1.** Time course of ERK induction after EGF stimulation and phosphoproteomic data for ERK1 and PLCG2.

**Figure supplement 1—source data 1.** Table showing phospho-ERK1/2 levels normalized to total-ERK1/2 levels (as quantified by measuring band intensities using ImageJ) in MDA-MB-468 cells treated with SHP099 and EGF as indicated.

**Figure supplement 1—source data 2.** Table showing peptide m/z values and TMT relative abundance values from MS3 spectrum of ERK1 phosphopeptide.

**Figure supplement 1—source data 3.** Table showing peptide m/z values and TMT relative abundance values from MS3 spectrum of PLCγ2 phosphopeptide.

**Figure supplement 2.** Principal component analysis.

**Figure supplement 3.** Global analysis of the SHP2 regulated phosphotyrosine proteome.

the absence of drug (condition 1), showing that SHP099 inhibition of SHP2 is rapidly reversible in MDA-MB-468 cells (*Figure 1A*, *Figure 1—figure supplement 1A*).

To obtain an in-depth view of the effects of SHP2 inhibition on EGFR signaling, we performed quantitative phosphoproteomics, monitoring dynamic changes in pY abundance as a function of time under DMSO, SHP099, and washout conditions (*Figure 1B*). Tryptic peptides from DMSO, SHP099, and SHP099-washout groups were enriched for phosphopeptides using immobilized metal affinity chromatography (IMAC) prior to labeling with 11-plex isobaric tandem mass tags (TMT). Tyrosine phosphorylated (pY) peptides were then immunoprecitiated from the TMT-labeled, pooled samples using an anti-pY antibody, and the recovered pY-containing peptides were analyzed by LC-MS3 mass spectrometry (*Supplementary files 1* and *2* and *Figure 1—figure supplement 1B,C*). Principal component analysis revealed a high degree of consistency between matched samples from the two biological replicates (*Figure 1—figure supplement 2*), which yielded relative quantification for several hundred pY-containing peptides, with good concordance between biological replicates (*Figure 1C* and *Supplementary files 1* and *2*). The Western blot (*Figure 1A*) and mass spectrometry data for the dynamics of pY204 of ERK1 (*Figure 1—figure supplement 1B*) and pY187 of ERK2 (*Supplementary files 1* and *2*) show the same pattern of response, confirming that key pY marks associated with EGF-induced signaling events were accurately determined in the TMT experiment.

The phosphoproteomic data revealed a spectrum of dependencies on EGF and SHP2 (*Figure 1—figure supplement 3A*). These were classified on the basis of both direction of change (increase or decrease) and timing. EGF dependencies were established based on responses to EGF addition in the absence of drug and were classified into five categories: fast, medium, or slow increases, neutral (no significant change), or decrease. The dependence of each phosphosite on SHP2 activity was established by assessing how phosphorylation levels were altered when SHP2 was inhibited: negative (more phosphorylation with inhibition; *Figure 1—figure supplement 3A*, top two rows), positive (less phosphorylation with inhibition; *Figure 1—figure supplement 3A*, bottom two rows), or neutral (*Figure 1—figure supplement 3A*, middle row). Positive and negative dependencies were further subdivided based on whether the change was present before EGF stimulation (*Figure 1—figure supplement 3A*, first and fourth rows) or only after (*Figure 1—figure supplement 3A*, second and fifth rows).

As anticipated, SHP2 inhibition resulted in increased abundance of pY marks at a number of sites (*Figure 1—figure supplement 3A*, top two rows), the expected effect of blocking a tyrosine phosphatase. Remarkably, however, we found an even larger family of pY sites that showed a decreased abundance when SHP2 is inhibited even at early time points (*Figure 1—figure supplement 3A*, bottom two rows), indicating that SHP2 inhibition had direct and/or indirect activities other than direct dephosphorylation of substrates. Additionally, SHP2 exhibited EGFR-independent regulation of a number of pY sites: a subset of sites was regulated (either positively or negatively) by SHP2 prior to EGF addition (*Figure 1—figure supplement 3A*, first and fourth rows), showing that SHP2 had a role in shaping the basal signaling state of these cells. Motif analysis of phosphosites associated with different classes of SHP2 responsiveness did not reveal enrichment for specific motifs (residues

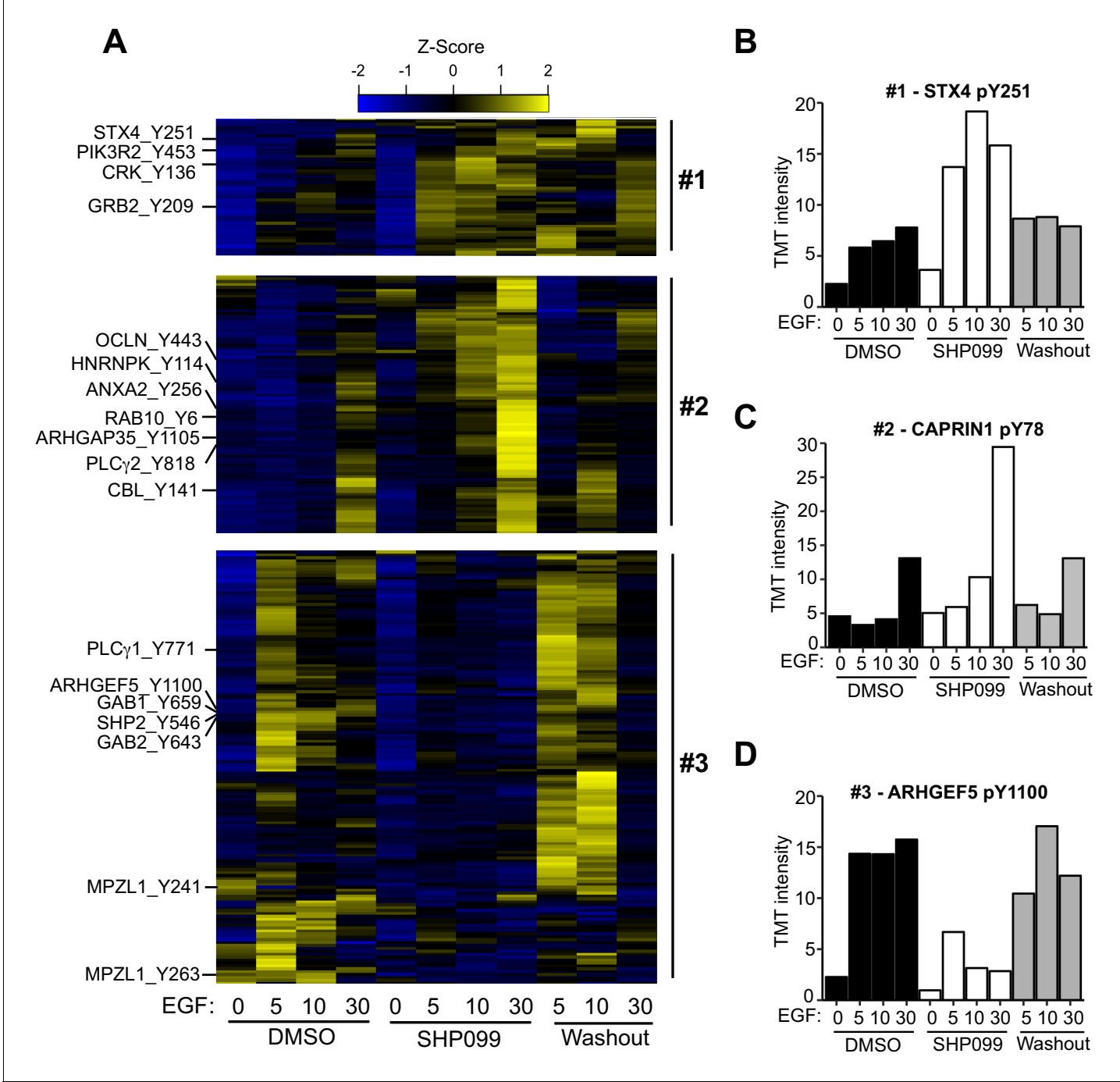

**Figure 2.** Quantitative phosphoproteomics reveals distinct dynamic responses to SHP2 inhibition. (**A**) Heatmap showing three classes of dynamic response in which inhibition of SHP2 modulates the effect of EGF stimulation on pY abundance. Specific examples from each cluster are indicated to the left of the heatmap. (**B**) Plot of pY abundance as a function of treatment condition for Y251 of STX4, an example of an early substrate-like response pattern to SHP2 inhibition. (**C**) Plot of pY abundance as a function of treatment condition for Y818 of PLCG2, an example of a late substrate-like response pattern to SHP2 inhibition. (**D**) Plot of pY abundance as a function of treatment condition for Y1100 of ARHGEF5, an example of a site where the abundance of the mark decreases when SHP2 is inhibited, and rebounds after compound washout.

The online version of this article includes the following source data and figure supplement(s) for figure 2:

**Source data 1.** Table showing TMT relative abundance values for Y251 of STX4.

**Source data 2.** Table showing TMT relative abundance values for Y78 of CAPRIN1.

**Source data 3.** Table showing TMT relative abundance values for Y1100 of ARHGEF5.

**Figure supplement 1.** Kinetic pattern of pY response to EGF stimulation under DMSO, SHP099, and washout conditions.

*Figure 2 continued on next page*

*Figure 2 continued*

**Figure supplement 1—source data 1.** Annotated heat maps for proteins in cluster 1.

**Figure supplement 1—source data 2.** Annotated heat maps for proteins in cluster 2.

**Figure supplement 1—source data 3.** Annotated heat maps for proteins in cluster 3.

**Figure supplement 1—source data 4.** Annotated heat maps for proteins in cluster 4.

**Figure supplement 1—source data 5.** Annotated heat maps for proteins in cluster 5.

**Figure supplement 1—source data 6.** Annotated heat maps for proteins in cluster 6.

**Figure supplement 1—source data 7.** Alphabetical list of proteins in cluster 1, with pY sites as indicated.

**Figure supplement 1—source data 8.** Alphabetical list of proteins in cluster 2, with pY sites as indicated.

**Figure supplement 1—source data 9.** Alphabetical list of proteins in cluster 3, with pY sites as indicated.

**Figure supplement 1—source data 10.** Alphabetical list of proteins in cluster 4, with pY sites as indicated.

**Figure supplement 1—source data 11.** Alphabetical list of proteins in cluster 5, with pY sites as indicated.

**Figure supplement 1—source data 12.** Alphabetical list of proteins in cluster 6, with pY sites as indicated.

**Figure supplement 2.** Classification of SHP099-responsive phosphotyrosine sites based on EGF temporal dynamics and function.

**Figure supplement 2—source data 1.** Table showing p-values and fold-changes in TMT relative abundance signal for all pY sites in untreated- or SHP099-treated-MDA-MB-468 cells after EGF stimulation for 5 min.

**Figure supplement 2—source data 2.** Table showing p-values and fold-changes in TMT relative abundance signal for all pY sites in untreated- or SHP099-treated-MDA-MB-468 cells after EGF stimulation for 10 min.

**Figure supplement 2—source data 3.** Table showing p-values and fold-changes in TMT relative abundance signal for all pY sites in untreated- or SHP099-treated-MDA-MB-468 cells after EGF stimulation for 30 min.

flanking the modified tyrosine residue; *Figure 1—figure supplement 3B*). Remarkably, sites from different regulatory classes could even occur together in the same protein (see below), highlighting the importance of considering specific sites of phosphorylation when evaluating the impact of a phosphatase on a protein substrate.

Hierarchical clustering was performed to identify groups of pY sites with similar kinetics in DMSO, SHP099, and washout conditions. Among the six kinetic profiles that emerged (*Figure 2—figure supplement 1*), we highlight three clusters of sites implicated in EGFR signaling and that display distinct responses to the drug (*Figure 2A*). The first class of responses, which includes the regulatory subunits of PI3K, occludin, Syntaxin 4 (*Figure 2B*), the adapter proteins CRK and GRB2, and ARHGAP35 (also known as p190RhoGAP), among other proteins, shows quantitatively increased tyrosine phosphorylation in the presence of SHP099 that rapidly disappears upon drug washout (*Figure 2A*), as predicted for a SHP2 substrate. The second class of sites accumulates pY marks slowly under SHP099 inhibition (at 30 min after EGF treatment), and not when SHP099 is omitted or washed out. This pattern, observed for pY sites on CBL E3 ligase, RAB10, hnRNPs, PLCγ proteins, and CAPRIN1 (*Figure 2C*), also matches the response predicted for a SHP2 substrate, but with a time delay, suggesting that the action of SHP2 on these proteins might require an intervening event, such as relocalization (e.g. after endocytosis of active EGFR signaling complexes), or alternatively, that the effect is indirect. The large number of potential substrates identified in this work suggests that SHP2 may catalyze dephosphorylation of many different proteins on different time scales.

The third pattern of response observed is one in which accumulation of pY in response to EGF *depends* on the release of SHP2 from inhibition by SHP099. Examples of sites that fall into this category include Y1100 of ARHGEF5 (*Figure 2D*), Y659 of GAB1, Y643 of GAB2, as well as Y546 and Y584 of SHP2 itself.

Functional classification of EGF-responsive pY sites (by GSEA and Reactome) differentially regulated by SHP099 (*Figure 2—figure supplement 2A–C*) reveals enrichment of six major cellular processes (*Figure 2—figure supplement 2D*). As expected, proteins implicated in MAPK and PI3K signaling, including EGFR and SHP2 itself, contain numerous EGF-responsive phosphosites differentially regulated by SHP099 treatment. The largest functional group includes proteins implicated in adhesion and migration, including tight junction proteins, catenins, Rho-GEF, and Rho-GAP proteins. These findings, along with data on the abundance of cytoskeletal proteins differentially affected by SHP099 (*Figure 2—figure supplement 2D*), are consistent with earlier results, suggesting that SHP2 promotes migration in MDA-MB-468 cells by regulating EGF-induced lamellipodia persistence (*Hartman et al., 2013*). Proteins implicated in transcription comprise another functional class with a number of differentially regulated phosphosites (*Figure 2—figure supplement 2D*). Two additional

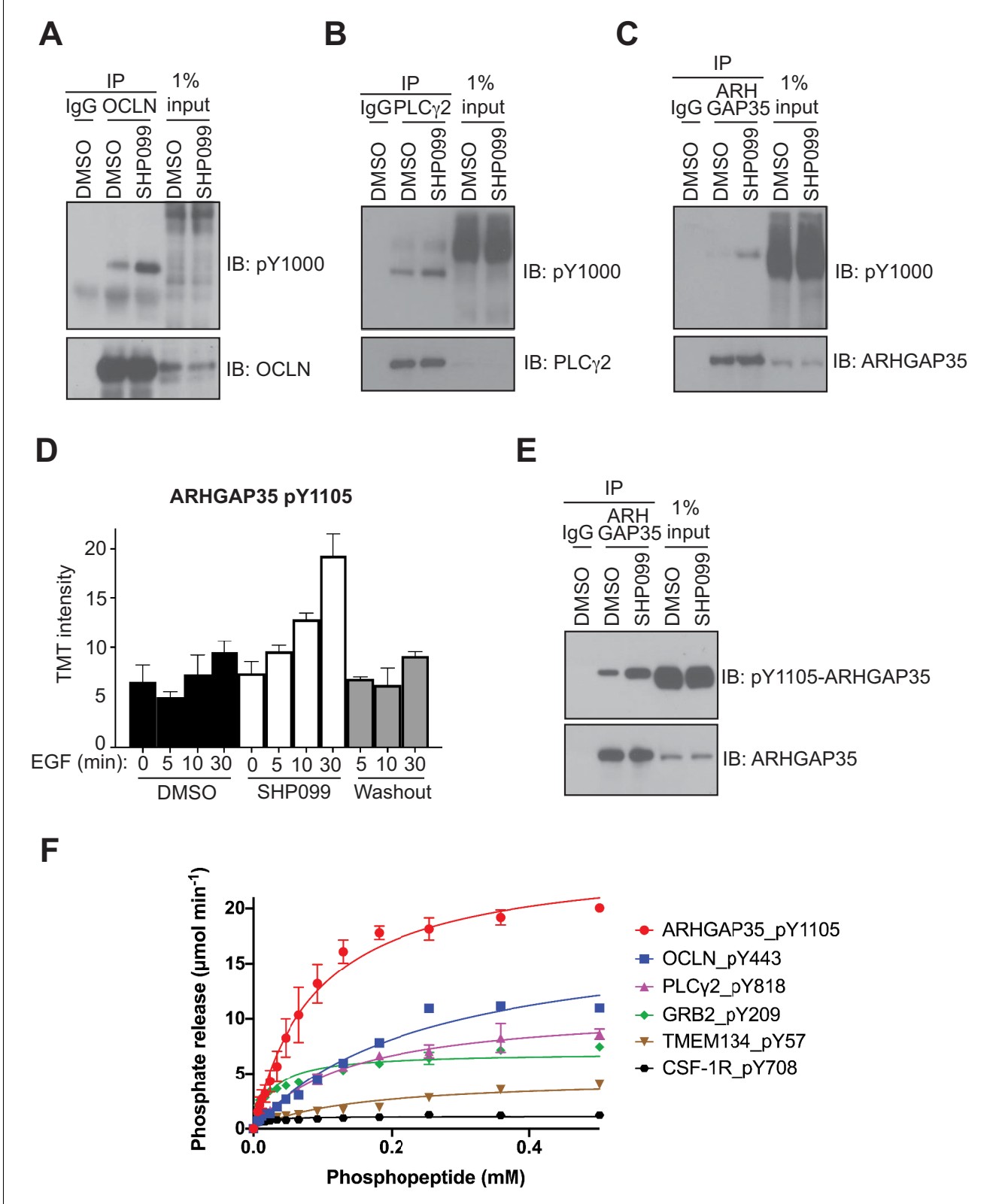

**Figure 3.** Evaluation of occludin, ARHGAP35, and PLCγ2 as SHP2 substrates. MDA-MB-468 cells pre-treated with DMSO carrier or SHP099 (10 μM) for 2 hr were mock treated or stimulated with EGF (10 nM) for 10 min (**A**) or 30 min (**B, C, and E**). Total cell lysates were immunoprecipitated with occludin (**A**), PLCγ2 (**B**), or ARHGAP35 (**C, E**) antibodies and the eluted samples were subjected to Western blotting with the phosphotyrosine antibody pY1000 (**A–C**), pY1105-ARHGAP35 (**E**), and anti-protein antibodies as indicated (**A–C, E**). Input samples represent 1% of total cell lysate. Non-specific IgG was

*Figure 3 continued on next page*

*Figure 3 continued*

used as a negative control. (D) Plot of pY abundance as a function of treatment condition for Y1105 of ARHGAP35. (F) Dephosphorylation activity of full-length wild-type SHP2, activated with 6 μM bisphosphorylated IRS-1 peptide [SLNY(p)IDLDLVKdPEG8-LSTY(p)ASINFQK], toward synthetic phosphopeptides (OCLN_pY443, PLCG2_pY818, GRB2_pY209, ARHGAP35_1105, TMEM134_pY57, CSF-1R_pY708). Immunoprecipitation – Western blot assays are representative of at least two independent biological replicates.

The online version of this article includes the following source data and figure supplement(s) for figure 3:

**Source data 1.** Table showing TMT relative abundance values for Y1105 of ARHGAP35.

**Source data 2.** Table showing micromoles of phosphate released per minute for all phosphopeptides tested for dephosphorylation by IRS1-activated wild-type SHP2.

**Figure supplement 1.** Evaluation of GRB2, occludin, and PLCγ2 as SHP2 substrates.

**Figure supplement 1—source data 1.** Table showing TMT relative abundance values for Y818 of PLCG2.

**Figure supplement 1—source data 2.** Table showing TMT relative abundance values for Y443 of OCLN.

**Figure supplement 1—source data 3.** Table showing TMT relative abundance values for Y209 of GRB2.

functional categories enriched in the analysis comprise endocytosis and mRNA processing, neither of which have previously been linked to SHP2 activity. Most of the sites in these two categories display a substrate-like response with SHP099 treatment. In addition, SHP099 also affected phosphorylation of CBL and CBLB, E3 ligases that ubiquitinate EGFR and facilitate its recruitment to clathrin-coated pits for endocytosis. Together, these results highlight the diversity of influence of SHP2, both in the function of proteins that it affects and in the manner in which it affects their phosphorylation patterns.

## Identification of new substrates of SHP2

Among proteins that show a substrate-like pattern, only one of the phosphosites observed has previously been reported to be dephosphorylated by SHP2 (GRB2 pY209; ARHGAP35 has also been reported to be a substrate, but its pY site has not been reported) (*Bregeon et al., 2009*; *Chardin et al., 1993*). Therefore, we carried out follow-up studies to determine whether any of the proteins with sites newly identified in this proteome-wide analysis are indeed directly dephosphorylated by SHP2.

Of the potential substrates identified from the mass spectrometry data, we identified 20 proteins with sites that (1) showed an increase upon EGF stimulation, (2) showed an increase of greater than twofold in the presence of SHP099 when compared to the DMSO control for at least one timepoint, (3) were not previously reported as SHP2 substrates, and (4) had commercial antibodies available for immunoprecipitation studies. Of these twenty proteins screened (*Supplementary file 3*), three that were reliably immunoprecipiated at endogenous abundance showed phosphosite enrichment after EGF stimulation in the presence of SHP099 as judged by Western blot with an anti-pY antibody. These three proteins are occludin, PLCγ2, and ARHGAGP35 (*Figure 3A–C*). Immunoprecipitation of pY-modified proteins with an anti-pY antibody in EGF-stimulated cells treated with SHP099 likewise showed that PLCγ2, occludin, and GRB2 accumulate pY in the presence of SHP099, as judged by Western blot (*Figure 3—figure supplement 1*). Because it was not possible to immunoprecipitate ARHGAP35 with either the anti-pY antibody or an antibody that specifically reognizes the pY1105 phosphosite, we immunoblotted anti-ARHGAP35 immunoprecipitates with the anti-pY1105 antibody to confirm that accumulation of this specific mark occurs when SHP2 is inhibited by SHP099 (*Figure 3D,E*). In addition, enzyme assays using purified SHP2 and the pY-containing peptides that contain the SHP099-enriched phsophosites identified in the proteomic studies confirm that SHP2 can robustly remove the phosphate mark from pY209 of GRB2, pY443 of occludin, pY1105 of ARHGAP35, and pY818 of PLCγ2, whereas neither a control peptide containing pY708 from CSF-1R nor a Class III peptide from TMEM134 that shows a protection-type pattern are robust SHP2 substrates (*Figure 3F*). Together, these data suggest that ARHGAP35, PLCγ2, and occludin are newly identified substrates for SHP2 in EGF-stimulated MDA-MB-468 cells.

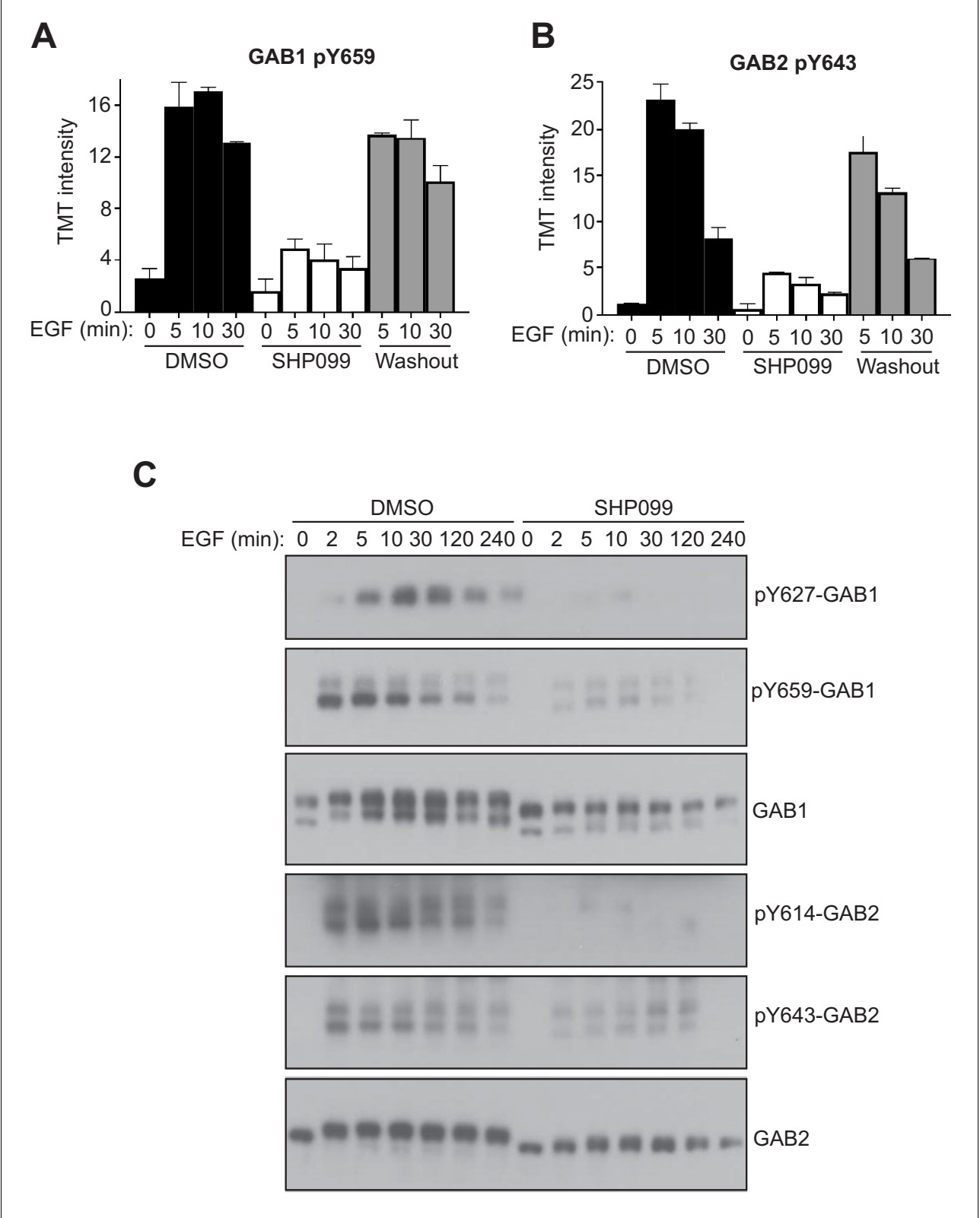

**Figure 4.** SHP2 inhibition results in reduced pY abundance at its interaction motifs. (A, B) TMT signal-to-noise intensities of GAB1 pY659 (A) and GAB2 pY643 (B) peptides showing dynamic changes in phosphorylation under DMSO- (solid line) and SHP099-treated (dotted line) conditions. (C) MDA-MB-468 cells pre-treated with DMSO carrier or SHP099 (10 µM) for 2 hr were mock treated or stimulated with EGF (10 nM). Total GAB1, total GAB2, GAB1

*Figure 4 continued on next page*

*Figure 4 continued*

pY659, and GAB2 pY643 were analyzed by Western blot with both anti-protein and phosphospecific antibodies as a function of time after EGF addition. Western blot results are representative of at least two independent biological replicates.

The online version of this article includes the following source data and figure supplement(s) for figure 4:

**Source data 1.** Table showing TMT relative abundance values for Y659 of GAB1.
**Source data 2.** Table showing TMT relative abundance values for Y643 of GAB2.
**Figure supplement 1.** SHP2 regulates GAB1 and GAB2 phosphorylation in multiple cellular contexts.

## Allosteric inhibition of SHP2 results in reduced pY abundance at its interaction motifs

SHP099 allosterically stabilizes the autoinhibited conformation of SHP2 (*Chen et al., 2016*), thereby not only inhibiting the catalytic activity of the enzyme, but also suppressing binding of its two SH2 domains to pY-containing motifs. Because the third pattern of response shows accumulation of pY after SHP2 is released from inhibition, we performed a dynamic analysis of whether SHP2 directly protects sites in this class from dephosphorylation and whether this protection is lost upon SHP099 binding by Western blot. Y659 of GAB1 and Y643 of GAB2 both show dramatic reductions in pY abundance upon SHP099 treatment and recover their pY marks upon compound washout (*Figure 4*), as judged by the phosphoproteomic data and confirmed across an extended time course by Western blot. These data are fully consistent with prior work indicating that the SH2 domains of SHP2 can bind to the bisphosphotyrosine-containing motifs of GAB1 (pY627/pY659) and GAB2 (pY614/pY643) to relieve SHP2 autoinhibition (*Arnaud et al., 2004*; *Cunnick et al., 2001*).

To determine whether pY sites of GAB1 and GAB2 exhibit the same pattern of response in other cellular contexts, we also treated other EGFR-driven cancer cell lines with SHP099. KYSE520, an EGFR-amplified esophageal cancer cell line, showed loss of GAB1 Y659 phosphorylation with SHP099 treatment under conditions of EGF stimulation (*Figure 4—figure supplement 1A*). H1975 cells, which carry an EGFR activating mutation, displayed high levels of constitutive GAB2-Y643 phosphorylation. SHP099 treatment eliminated phosphorylation of this site, and washout of the drug restores the mark within 5 min (*Figure 4—figure supplement 1B*). In addition, we tested whether deletion of the SHP2 gene recapitulated the effects of chemical inhibition on the phosphorylation of Y659 of GAB1. For these studies, we used a *PTPN11* knockout U2OS cell line prepared by CRISPR/Cas9 mediated genome editing (*LaRochelle et al., 2018*). When stimulated with EGF, SHP2-null U2OS cells showed a reduction in GAB1 pY659 levels when compared to parental cells, and reintroduction of SHP2 fully rescued the level of accumulated pY659 (*Figure 4—figure supplement 1C*). Stimulation of U2OS cells with PDGFββ also showed an induction of the GAB1 pY659 mark that is lost upon treatment with SHP099 (*Figure 4—figure supplement 1D*). Similarly, stimulation of the T cell receptor on Jurkat cells with an anti-CD3 antibody induced GAB1 and GAB2 phosphorylation, and this induction was attenuated by SHP099 treatment (*Figure 4—figure supplement 1E*). These findings show that the effect of SHP2 on the abundance of GAB1 and GAB2 pY marks is broadly shared among a range of growth factor and antigen receptor signaling systems.

To determine how inhibition of SHP2 reduced pY abundance at these sites, we generated a set of site-specific SHP2 mutants and asked whether they could rescue GAB1-pY659 phosphorylation in U2OS *PTPN11* null cells. We constructed point mutations that abolish autoinhibition (E76K), eliminate catalytic activity (C459E), or disable both autoinhibition and catalytic activity (E76K/C459E). We also created protein truncations that contain only the catalytic domain (PTP) or only the SH2 domains (SH2) (*Figure 5A*). When stimulated with EGF, C459E, E76K, and the E76K/C459E double mutant, all restored GAB1-pY659 phosphorylation to a level similar to that of cells rescued with wild-type SHP2 (*Figure 5B*). When the SH2 or PTP constructs were introduced into U2OS cells lacking endogenous SHP2, the PTP fragment failed to rescue, whereas the SH2 fragment was as active as wild-type SHP2 (*Figure 5C*).

Since MDA-MB-468 cells cannot survive without SHP2 (see also the gene essentiality data from avana_public_18Q2 library, Project Achilles; *Meyers et al., 2017*), we stably expressed the SH2 or PTP fragments in parental MDA-MB-468 cells and chemically inhibited endogenous SHP2. These cell lines were stimulated with EGF and probed for phospho-GAB1 and phospho-GAB2. As expected, SHP099 treatment significantly reduced the abundance of pY659 on GAB1, as well as that of pY643

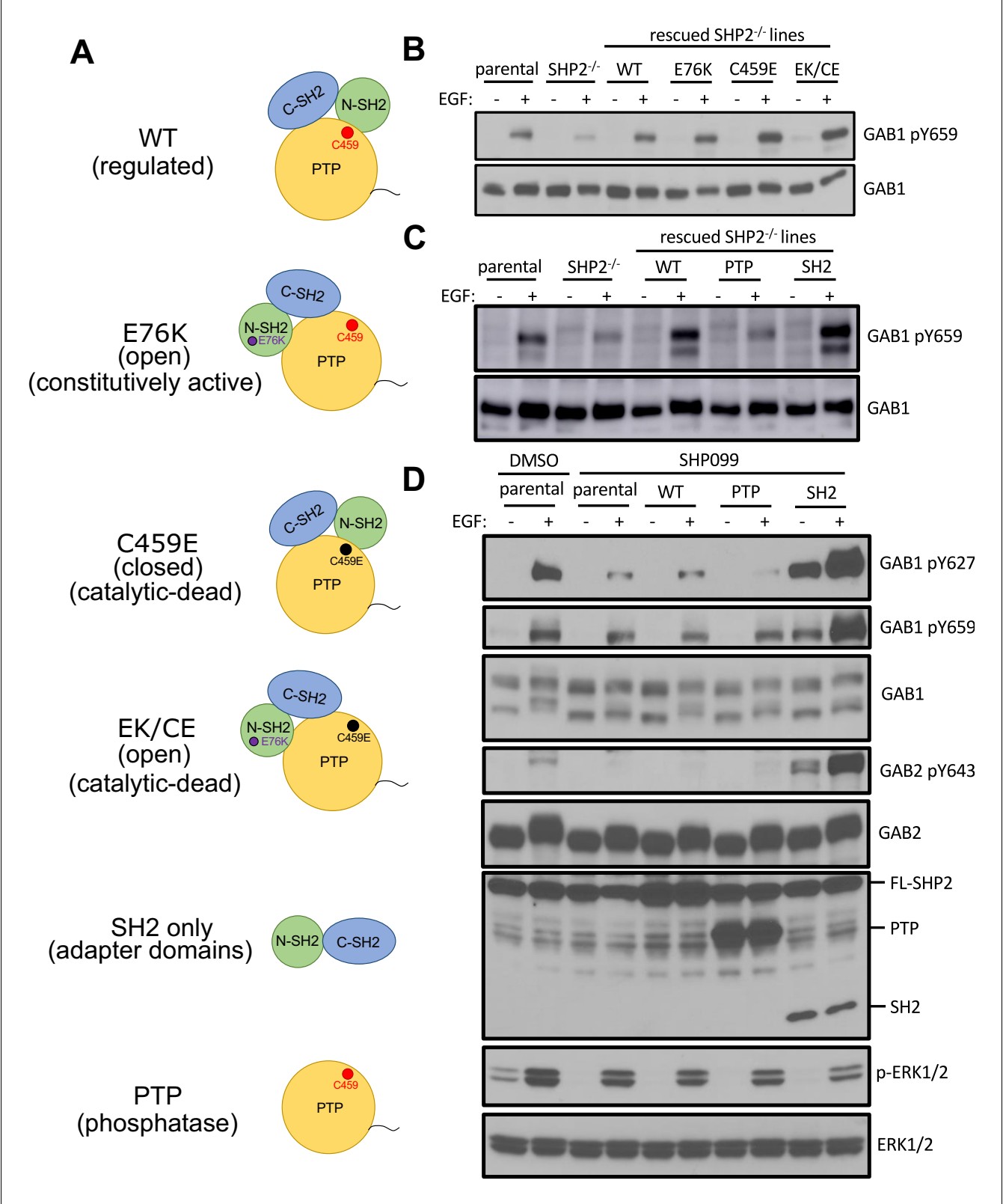

**Figure 5.** The SH2 domains of SHP2, but not its PTP domain, protect specific GAB1 and GAB2 pY marks. (**A**) Schematic representation of wild-type, point mutants, and deletion constructs of SHP2 tested. (**B, C**) Parental-, SHP2 knockout-, or SHP2 knockout U2OS cells stably expressing various SHP2 'rescue' constructs were cultured with or without EGF stimulation (10 nM) for 10 min. Cells were lysed and subjected to Western blotting using anti-pY659-GAB1 and anti-GAB1 antibodies. (**D**) MDA-MB-468 cells stably expressing wild-type, PTP, or SH2 domains pre-treated with SHP099 were

*Figure 5 continued on next page*

*Figure 5 continued*

stimulated with EGF and immunoblotted after lysis using the specified antibodies. Western blot results are representative of at least n = 2 independent biological replicates.

The online version of this article includes the following source data and figure supplement(s) for figure 5:

**Figure supplement 1.** SHP2 protects MPZL1 phosphotyrosine residues.
**Figure supplement 1—source data 1.** Table showing TMT relative abundance values for Y263 of MPZL1.
**Figure supplement 1—source data 2.** Table showing TMT relative abundance values for Y241 of MPZL1.
**Figure supplement 2.** SHP2 does not affect the deposition of phosphate marks on GAB1/2 by SFKs.
**Figure supplement 3.** SHP2 regulates GAB1 association with activated EGFR.
**Figure supplement 3—source data 1.** Table showing quantified EGFR-GAB1 PLA signal in MDA-MB-468 cells with SHP099 and EGF as indicated.

on GAB2. GAB1 pY627, which has been reported to bind to the N-SH2 domain of SHP2 (*Cunnick et al., 2001*), also showed reduced phosphorylation. Consistent with the results from U2OS cells, the SH2 fragment, but not the PTP fragment was active in increasing pY abundance at these positions of GAB1 and GAB2 (*Figure 5D*). Indeed, the SH2 fragment stabilizes these GAB1 and GAB2 phospho marks even in the absence of EGF stimulation (*Figure 5D*), consistent with a dominant protective function that is independent of the catalytic activity of the enzyme. In addition, the SHP099-resistant variant of SHP2 (T253M/Q257L) rescues protection of the GAB1 and GAB2 pY sites (*Figure 6A,B*), and point mutation of the key arginine residue of each SH2 domain (R32M of the N-SH2 domain or R138M of the C-SH2 domain) suppresses phosphosite protection of these sites, as does the R32M/R138M double mutant (*Figure 6C,D*). Together, these findings show that the catalytic activity of SHP2 is not required for the protection of the pY659 and pY643 marks on GAB1 and GAB2, respectively, and argue that the presence of tandem SH2 domains in SHP2 is needed to bind and protect pYs on GAB1 and GAB2 from phosphatase-mediated dephosphorylation.

To assess whether this protection mechanism applies to other SHP2 interacting proteins that show reduced pY abundance in the presence of SHP099, we studied MPZL1, a cell surface receptor involved in extracellular matrix-induced signaling (*Beigbeder et al., 2017*; *Zhao et al., 2002*). MPZL1 contains an immunoreceptor tyrosine-based inhibitory motif (ITIM) that, when doubly phosphorylated at Y241 and Y263, can bind the tandem SH2 domains of SHP2 (*Zhao and Zhao, 2000*). Our phosphoproteomic data show that basal pY abundance at these sites is greatly reduced by SHP099 treatment and that the dependence of pY modifications on SHP2 at these two sites occurs independent of EGF (*Figure 5—figure supplement 1A,B*). Using immunoblot assays, we found that the tandem SH2 domains alone are also sufficient to protect the phosphate groups on Y241 and Y263 of MPZL1 (*Figure 5—figure supplement 1C*). It is also notable that the two pY sites on the SHP2 C-terminal tail exhibit a protection-type pattern in the presence of SHP099. Because the allosteric inhibitor locks SHP2 in the closed conformation, we speculate that the inhibitor interferes with membrane recruitment of SHP2 by preventing the SH2 domains from binding to pY residues deposited at membrane proximal sites by activated kinases and that removal of inhibitor permits membrane recruitment to sites of EGFR activation, phosphorylation of the SHP2 C-terminal tail, and subsequent GRB2 recruitment.

Multiple studies have reported that GAB1, GAB2, and MPZL1 can be phosphorylated by Src family kinases (SFKs) (*Chan et al., 2003*; *Kong et al., 2003*; *Kusano et al., 2008*). Indeed, MDA-MB-468 cells treated with an SFK inhibitor (saracatinib/AZD0530) showed greatly reduced modification of GAB1-Y659 and GAB2-Y643 (*Figure 5—figure supplement 2A*), suggesting that SFK activity is required to phosphorylate the SHP2 binding sites on GAB1 and GAB2. SHP099, however, had no detectable effect on the abundance of pY at Y416 of the activation loop of SFKs (*Figure 5—figure supplement 2B*), or on the abundance of pY at Y527, which maintains SFKs in their autoinhibitory conformation (*Roskoski, 2005*). Although it is possible that SFK activity in the GAB complex is not reflected by assessment of bulk SFK activity in the whole-ell lysate, when combined with the phosphosite protection results from the studies with various forms of SHP2 above, these data suggest that SHP2 acts to increase the half-life of pY modifications at sites on GAB1 and GAB2 that are phosphorylated by SFKs and then bound by SHP2.

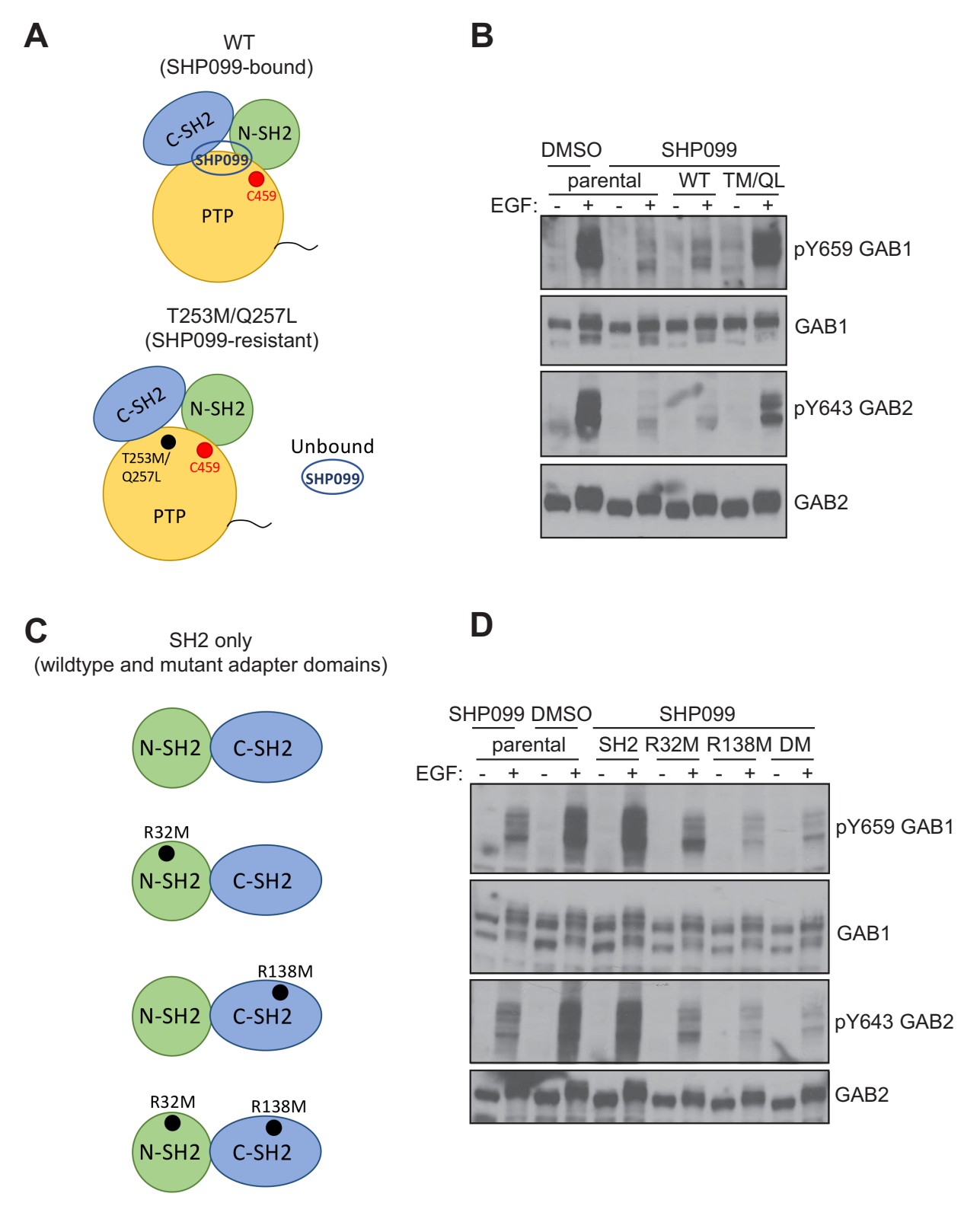

**Figure 6.** Effects of SHP099-resistance and pY binding-site mutations on pY abundance at protected sites on GAB1 and GAB2. (**A**) Schematic illustrating wild-type and SHP099-resistant forms of SHP2. (**B**) Parental MDA-MB-468 cells or MDA-MB-468 cells stably expressing wild-type or T253M/Q257L SHP2 proteins were pre-treated with SHP099 (10 μM). The cells were then mock treated or stimulated with EGF (10 nM) for 10 min and immunoblotted after lysis using the specified antibodies. (**C**) Schematic illustrating wild-type and mutated forms of SH2-only tandem fragments of

*Figure 6 continued on next page*

*Figure 6 continued*

SHP2. (D) Parental MDA-MB-468 cells or MDA-MB-468 cells stably expressing wild-type or mutated tandem SH2 fragments of SHP2 (SH2) as indicated (R32M, R138M, and R32M/R138M, denoted DM here) were pre-treated with SHP099 (10 μM). The cells were then mock treated or stimulated with EGF (10 nM) for 10 min and immunoblotted after lysis using the specified antibodies. Western blot results are representative of at least n = 2 independent biological replicates.

## SHP2 is required for membrane localization of GAB1

To determine whether activation of SHP2 is required for recruitment of GAB1 to EGFR at the plasma membrane, we used a proximity ligation assay (PLA). MDA-MB-468 cells were mock treated or stimulated with EGF in the presence of DMSO or SHP099, fixed in paraformaldehyde, and probed for co-localization of GAB1 and EGFR. Under serum-starved (basal) conditions, the PLA signal was minimal and unaffected by SHP2 inhibition with SHP099. In contrast, within 2 min of EGF addition, cells treated with DMSO alone displayed a strong PLA signal indicative of GAB1 and EGFR co-localization, whereas SHP099 treatment reduced the number of binding events by 85% (p<0.05; *Figure 5— figure supplement 3*). These results suggest that SHP2 is required to assemble the EGFR-GAB1 signaling hub crucial for MAPK and PI3K signaling.

## Discussion

Here, we performed quantitative phosphoproteomics to determine how the responses of cells to EGFR stimulation are modulated by inhibition of SHP2. We identified over 400 pY sites that exhibit increases or decreases in phosphorylation in response to SHP099 addition and washout. The dynamics of these changes in pY abundance reveal six distinct response signatures that provide a number of insights into how cancer cells depend on SHP2 in RTK signaling. Three of the six patterns identified by hierarchical clustering correspond to EGFR-dependent pY abundance changes that are also modulated by SHP2. These patterns include an early SHP2 substrate-like pattern (early responders that increase in abundance upon SHP2 inhibition), a late SHP2 substrate-like pattern (late responders also increased upon SH2 inhibition), and a pattern in which pY levels are unexpectedly decreased when SHP2 is inhibited (protected sites).

Among proteins that exhibit a substrate-like pattern, occludin, ARHGAP35, and PLCγ2 were also confirmed to have specific pYs that are SHP2 substrate sites, opening up new avenues for future study of the downstream effects of SHP2 on cell migration and adhesion. A number of other early responder sites lie on proteins that transmit information from activated RTKs to Ras/MAPK and PI3K signaling cascades. For example, the pY209 mark on the adaptor protein Grb2, which inhibits Sos binding and downstream Ras activation (*Chardin et al., 1993*; *Li et al., 2001*), rapidly accumulates upon SHP2 inhibition, suggesting that one important role of SHP2 in these cells is to remove an inhibitory mark and promote Grb2-dependent protein–protein interactions that drive immediate early signaling. Similarly, the adapter Crk also has two early responder pY sites at positions 108 and 136 that accumulate upon SHP2 inhibition; their removal by SHP2 may promote the ability of Crk to propagate the EGFR signal. Likewise, the PI3K regulatory subunits PI3KR1 and PI3KR2 accumulate pY marks in their inter-SH2 (iSH2) domains at analogous positions between residues 452–470 that may suppress effector function when SHP2 is inhibited, and PI3KR3 also has a pY residue in its iSH2 domain that also shows a substrate-like response at later timepoints. Not surprisingly, SHP2 inhibition also affects the pY abundance on numerous proteins involved in cell adhesion and migration, as well as on cytoskeletal components (*Figure 2—figure supplement 2D*).

Our data also reveal many proteins with SHP2-dependent changes in pY abundance that function in cellular processes previously unlinked to SHP2. These proteins include effectors of distal signaling responses such as phospholipase C enzymes (PLCγ1 and PLCγ2). Also noteworthy are the late-response, substrate-like patterns that predominate among proteins implicated in mRNA processing and endocytosis factors (hnRNPA2B1, hnRNPA3, EPN1/2, RAB10, FNBP1L, ITSN2, STX4) (*Figure 2— figure supplement 2D*). Dissecting the role of SHP2 in these events should be a fertile area of future investigation.

Strikingly, the most frequent change in pY pattern following SHP2 inhibition is not substrate-like, but rather one in which pY levels fall. Others have also recently reported that binding of SHP2 can prevent dephosphorylation of certain pY sites in the response to stimulation of other RTKs

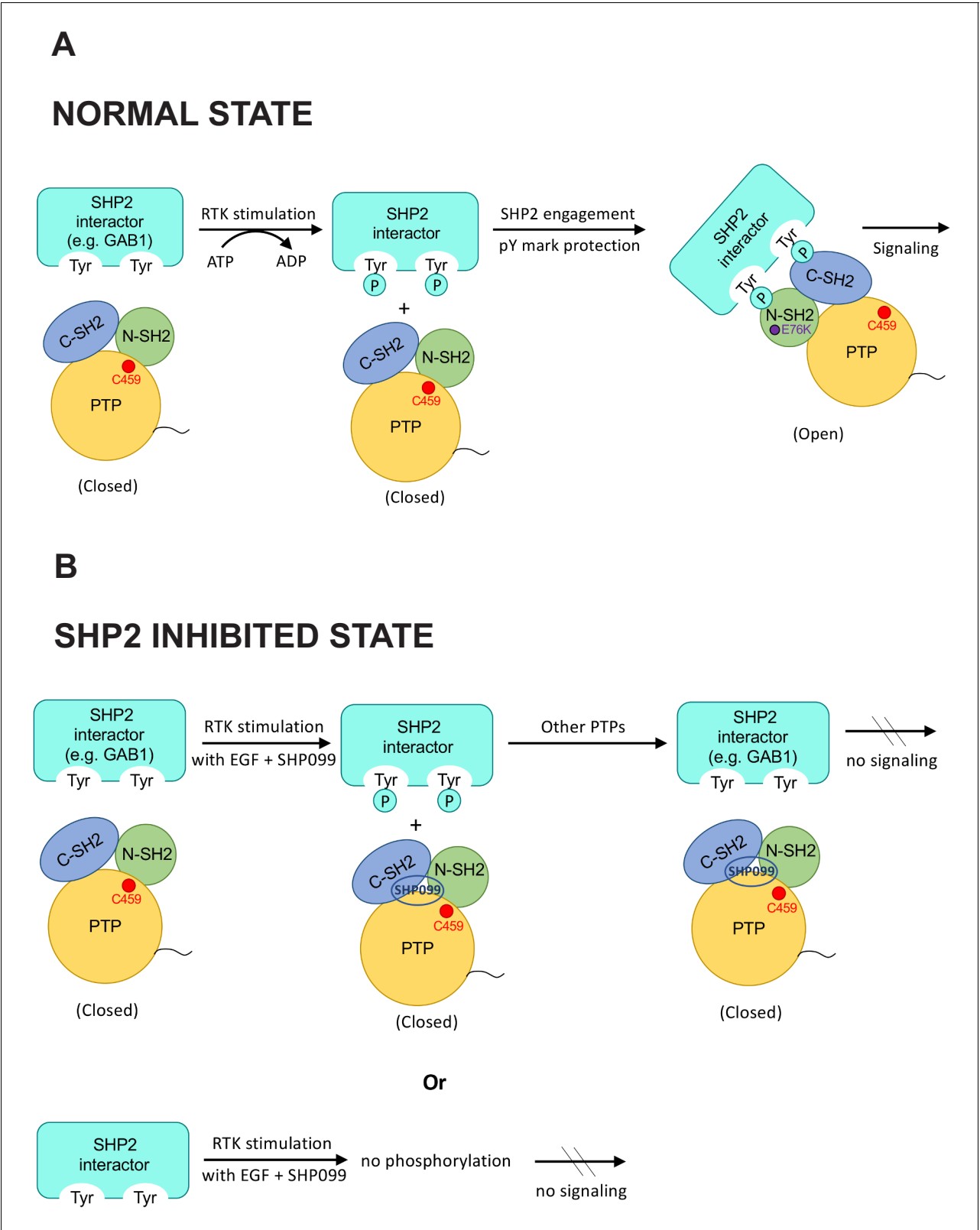

**Figure 7.** Model highlighting coordinated scaffolding and catalytic activities of SHP2. (**A**) Schematic model showing pY mark protection by stable engagement of a partner protein (e.g. GAB1) with the tandem SH2 domains of SHP2. (**B**) In the presence of SHP099, pY marks on putative SHP2 interactors are vulnerable to dephosphorylation, and SHP099 is not recruited to sites of RTK signaling. It is also possible that deposition of certain phosphate marks is SHP2 dependent (lower panel).

*Figure 7 continued on next page*

*Figure 7 continued*

The online version of this article includes the following source data and figure supplement(s) for figure 7:

**Figure supplement 1.** Proteins with SHP2-protected pY sites are enriched for multisite protein phosphorylation.

**Figure supplement 1—source data 1.** Table showing TMT relative abundance values for all SHP2-protected pY sites displaying multisite protein phosphorylation dynamics.

(*Batth et al., 2018*). This pattern of response is seen at pY residues on proteins recruited to the EGFR signaling hub, such as GAB1, GAB2, as well as on EGFR and SHP2 itself. To better understand this mode of pY regulation, we studied SHP2-dependent protection of pY sites on GAB1 and GAB2, adapters that induce SHP2 binding and activation upon tyrosine phosphorylation (*Arnaud et al., 2004*; *Crouin et al., 2001*; *Cunnick et al., 2001*). Binding and mutagenesis studies clearly demonstrate that the tandem SH2 domains of SHP2 shield pY binding sites on GAB1 and GAB2 from dephosphorylation. Also intriguing is the SHP2-mediated protection of nearby pairs of pY residues on several proteins, including MPZL1-pY241/pY263, a fibronectin-activated adhesion protein that can assemble into complexes containing SHP2 and Grb2 in a pY-dependent manner (*Figure 5—figure supplement 1* and *Beigbeder et al., 2017*). In fact, decreased phosphoabundace is observed on multiple proteins harboring nearby tyrosine residues (*Figure 7—figure supplement 1*), all of which may experience protection of their pY marks by the tandem SH2 domains of SHP2. The protection of pY sites by SHP2 highlights the scaffolding role of SHP2 in stimulating pY engagement by SH2 domains at sites of RTK activation (*Figure 7*). SHP2 scaffolding activity is particularly relevant to Noonan and LEOPARD syndromes, both of which result from missense SHP2 mutations. Though distinct, the two syndromes have overlapping clinical features. Mutations found in Noonan syndrome destabilize the autoinhibited form of the enzyme without substantially affecting the intrinsic catalytic activity of the phosphatase domain, leading to increased basal phosphatase activity, elevated scaffolding activity, and downstream induction of ERK (*Araki et al., 2004*; *Niihori et al., 2005*). LEOPARD syndrome mutations also destabilize the autoinhibited conformation of the enzyme, enhancing scaffolding activity, but in addition cripple the catalytic activity of the phosphatase domain. The increased preference for the open conformation in both Noonan and LEOPARD syndrome mutations promotes the protective 'arm' of SHP2 signaling, whereas only Noonan syndrome mutations retain the catalytic activity needed to drive the substrate-dependent features of signaling. The work reported here shows how central the scaffolding function is at the EGFR signaling hub and in the engagement of pY sites on GAB1 and GAB2 at the site of activation, with the phosphatase-dependent events, through adapter proteins like Grb2 and key signaling nodes like PI3K, required for full propagation of a growth factor induced signal to downstream effectors.

## Materials and methods

### Key resources table

| Reagent type (species) or resource | Designation | Source or reference | Identifiers | Additional information |
|---|---|---|---|---|
| Recombinant DNA reagent | pCMV-SHP2-WT | Addgene | RRID:Addgene_8381, plasmid # 8381 | |
| Recombinant DNA reagent | migR1-IRES-GFP | Addgene | RRID:Addgene_27490, plasmid # 27490 | |
| Chemical compound, drug | SHP2 inhibitor (SHP099) | DC chemicals | Catalog # DC9737 | |
| Chemical compound, drug | Src inhibitor (AZD0530) | Dr. Nathanael Gray (Dana-Farber Cancer Institute) | | |
| Antibody | Mouse monoclonal CD3 (UCHT1) | Thermo Fisher Scientific | RRID:AB_468857; Catalog # 16-0038-85 | 10 µg/ml for TCR stimulation of Jurkat cells |
| Antibody | Rabbit monoclonal Phospho-Tyr-1000 | Cell Signaling Technology | RRID:AB_2687925; Catalog # 8954 | Western blot (1:2000); Immunoprecipitation (1:100) |

*Continued on next page*

*Continued*

| Reagent type (species) or resource | Designation | Source or reference | Identifiers | Additional information |
|---|---|---|---|---|
| Antibody | Rabbit polyclonal Phospho-Thr202/Tyr204-Erk1/2 | Cell Signaling Technology | RRID:AB_331646; Catalog # 9101 | Western blot (1:2000) |
| Antibody | Rabbit polyclonal Erk1/2 | Cell Signaling Technology | RRID:AB_330744; Catalog # 9102 | Western blot (1:2000) |
| Antibody | Rabbit polyclonal GAB1-pY659 | Cell Signaling Technology | RRID:AB_2798014; Catalog # 12745 | Western blot (1:1000) |
| Antibody | Rabbit polyclonal GAB1-pY627 | Cell Signaling Technology | RRID:AB_2305002; Catalog # 3231 | Western blot (1:500) |
| Antibody | Rabbit polyclonal GAB2-pY643 | Thermo Fisher Scientific | RRID:AB_2554386; Catalog # PA5-37778 | Western blot (1:1000) |
| Antibody | Rabbit polyclonal GAB1 | Cell Signaling Technology | RRID:AB_2304999; Catalog # 3232 | Western blot (1:2000) |
| Antibody | Rabbit monoclonal GAB2 | Cell Signaling Technology | RRID:AB_10698601; Catalog # 3239 | Western blot (1:2000) |
| Antibody | Rabbit polyclonal pY263-MPZL1 | Cell Signaling Technology | RRID:AB_10715080; Catalog # 5543 | Western blot (1:1000) |
| Antibody | Rabbit monoclonal pY241-MPZL1 | Cell Signaling Technology | RRID:AB_10891793; Catalog # 8131 | Western blot (1:1000) |
| Antibody | Rabbit polyclonal MPZL1 | Cell Signaling Technology | RRID:AB_1904189; Catalog # 4157 | Western blot (1:2000) |
| Antibody | Rabbit polyclonal pY1105-ARHGAP35 | Thermo Fisher Scientific | RRID:AB_2553676; Catalog # PA5-36713 | Western blot (1:1000) |
| Antibody | Rabbit polyclonal ARHGAP35 | Cell Signaling Technology | RRID:AB_2115740; Catalog # 2562 | Western blot (1:2000); Immunoprecipitation (1:100) |
| Antibody | Rabbit monoclonal SHP2 | Cell Signaling Technology | RRID:AB_2174959; Catalog # 3397 | Western blot (1:500) |
| Antibody | Rabbit polyclonal SHP2 | Bethyl | RRID:AB_1040019; Catalog # A301-544A | Western blot (1:1000) |
| Antibody | Mouse monoclonal SHP2 | Santa Cruz | RRID:AB_628252; Catalog # sc-7384 | Immunoprecipitation (2 µg antibody per 500 µg cell lysate) |
| Antibody | Mouse monoclonal PLCG2 | Santa Cruz | RRID:AB_628120; Catalog # sc-5283 | Western blot (1:500); Immunoprecipitation (2 µg antibody per 500 µg cell lysate) |
| Antibody | Rabbit polyclonal OCLN | Bethyl | RRID:AB_2631690; Catalog # A305-297A | Western blot (1:1000); Immunoprecipitation (5 µg antibody per 1 mg cell lysate) |
| Antibody | Mouse monoclonal GRB2 | BD Transduction Laboratories | RRID:AB_397518; Catalog # 610112 | Western blot (1:2000) |
| Cell line (*H. sapiens*) | MDA-MB-468 | ATCC | RRID:CVCL_0419; HTB-132 | |
| Cell line (*H. sapiens*) | KYSE-520 | DSMZ | RRID:CVCL_1355; ACC 371 | |
| Cell line (*H. sapiens*) | U2OS | ATCC | RRID:CVCL_0042; HTB-96 | |
| Cell line (*H. sapiens*) | Jurkat | ATCC | RRID:CVCL_0367; TIB-152 | |
| Cell line (*H. sapiens*) | NCI-H1975 | ATCC | RRID:CVCL_UE30; CRL-5908 | |
| Cell line (*H. sapiens*) | SHP2 knockout U2OS cell line | *LaRochelle et al., 2018* | | |

*Continued on next page*

*Continued*

| Reagent type (species) or resource | Designation | Source or reference | Identifiers | Additional information |
|---|---|---|---|---|
| Peptide, recombinant protein | Human EGF | Gibco | PHG0311 | |
| Peptide, recombinant protein | Human PDGF-BB | Peprotech | Catalog # 10771–922 | |
| Peptide, recombinant protein | IRS1_pY1172–PEG–pY1222 peptide | This paper | | SLN{pY}IDLDLVK–dPEG8–LST{pY}ASINFQK; (custom synthesized by Genscript) |
| Peptide, recombinant protein | OCLN_pY443 peptide | This paper | | TGLQE{pY}KSLQS; (custom synthesized by Genscript) |
| Peptide, recombinant protein | PLCG2_pY818 peptide | This paper | | TRIQQ{pY}FPSNY (custom synthesized by Genscript) |
| Peptide, recombinant protein | GRB2_pY209 peptide | This paper | | MFPRN{pY}VTPVN; (custom synthesized by Genscript) |
| Peptide, recombinant protein | ARHGAP35_pY1105 peptide | This paper | | EENI{pY}SVPHDS; (custom synthesized by Genscript) |
| Peptide, recombinant protein | CSF-1R_pY708 peptide | This paper | | IHLEKK{pY}VRRDSGF; (custom synthesized by Genscript) |
| Peptide, recombinant protein | TMEM134_pY57 peptide | This paper | | KQSRLR{pY}QNLEND; (custom synthesized by Genscript) |
| Peptide, recombinant protein | Lysyl Endopeptidase, Mass Spectrometry Grade (Lys-C) | FUJIFILM Wako Pure Chemical Corporation | Catalog # 125–05061 | Protein cleavage for mass spectrometry (10 µg per 1 mg total protein) |
| Peptide, recombinant protein | Sequencing Grade Modified Trypsin | Promega | Catalog # V511C | Protein cleavage for mass spectrometry (10 µg per 1 mg total protein) |
| Peptide, recombinant protein | Wild-type SHP2 purified protein | *LaRochelle et al., 2016* | | |
| Commercial assay or kit | QuikChange II Site-Directed Mutagenesis kit | Agilent | Catalog # 200523 | |
| Commercial assay or kit | BCA protein assay | Pierce | Catalog # 23225 | |
| Commercial assay or kit | Malachite Green assay kit | Sigma–Aldrich | Catalog # MAK307 | |
| Commercial assay or kit | PLA kit | Sigma–Aldrich | Catalog # DUO92014 | |
| Other | NTA Magnetic Agarose beads | Qiagen | Catalog # 36113 | |
| Other | TMT 11-plex reagent | Thermo Scientific | Catalog # A37725 | |
| Software, algorithm | GraphPad Prism 9.0 | N/A | RRID:SCR_002798; https://www.graphpad.com/scientific-software/prism/ | |
| Software, algorithm | R 3.6.3 | N/A | RRID:SCR_001905; https://www.r-project.org/ | |
| Software, algorithm | CellProfiler | N/A | RRID:SCR_007358; https://cellprofiler.org/ | |

*Continued on next page*

*Continued*

| Reagent type (species) or resource | Designation | Source or reference | Identifiers | Additional information |
|---|---|---|---|---|
| Software, algorithm | Reactome database | N/A | RRID:SCR_003485; http://software.broadinstitute.org/gsea/msigdb/annotate.jsp | |
| Software, algorithm | Gene set enrichment analysis | *Mootha et al., 2003*; *Subramanian et al., 2005* | RRID:SCR_003199 | |
| Software, algorithm | Hierarchical clustering | MATLAB | RRID:SCR_001622; https://www.mathworks.com/products/matlab.html | |
| Software, algorithm | Principal component analysis | Python | RRID:SCR_008394; https://www.python.org/ | |
| Software, algorithm | Dependence classification | This paper | | See 'Dependence classification of phosphorylation site time courses' in Methods |
| Software, algorithm | Logo analysis | *Crooks, 2014* | RRID:SCR_010236 | |

## Plasmids, compounds, and ligands

Constructs used to rescue SHP2 knockout U2OS cells were obtained by cloning the SHP2 cDNA from pCMV-SHP2-WT (RRID: Addgene_8381) into the migR1-IRES-GFP vector (RRID:Addgene_27490). Point mutants (E76K, C459E, and DM) were generated from the parental SHP2-migR1 construct using site-directed mutagenesis (QuikChange II, Agilent, Catalog # 200523). SH2-only and phosphatase-only mutants were generated by cloning the PTP (220–525 aa) and N-SH2-C-SH2 (1–219 aa) fragments, respectively, into the migR1-IRES-GFP vector. SHP099 was obtained commercially from DC chemicals (Catalog # DC9737). AZD0530 (Saracatinib) was a gift from Dr. Nathanael Gray (Dana-Farber Cancer Institute). The CD3 monoclonal antibody (UCHT1) was purchased from Thermo Fisher Scientific (RRID:AB_468857). Recombinant human EGF and PDGF-BB were purchased from Gibco (Catalog # PHG0311) and Peprotech (Catalog # 10771–922), respectively.

## Antibodies

Antibodies used in this study were obtained commercially from the following sources: Phospho-Tyr-1000 (RRID:AB_2687925), Phospho-Thr202/Tyr204-Erk1/2 (RRID:AB_331646), Erk1/2 (RRID:AB_330744), GAB1-pY659 (RRID:AB_2798014), GAB1-pY627 (RRID:AB_2305002), GAB2-pY643 (RRID:AB_2554386), GAB1 (RRID:AB_2304999), GAB2 (RRID:AB_10698601), pY263-MPZL1 (RRID:AB_10715080), pY241-MPZL1 (RRID:AB_10891793), MPZL1 (RRID:AB_1904189), pY1105-ARHGAP35 (RRID:AB_2553676), ARHGAP35 (RRID:AB_2115740), SHP2 (RRID:AB_2174959, RRID:AB_1040019, RRID:AB_628252), PLCG2 (RRID:AB_628120), OCLN (RRID:AB_2631690), and GRB2 (RRID:AB_397518).

## Cell culture and generation of stable cell lines

All the parental cell lines used in this study (MDA-MB-468 [RRID:CVCL_0419], KYSE520 [RRID:CVCL_1355], U2OS [RRID:CVCL_0042], Jurkat [RRID:CVCL_0367], and NCI-H1975 [RRID:CVCL_UE30]) were purchased from ATCC. The SHP2 knockout U2OS cell line was created previously (*LaRochelle et al., 2018*). To generate rescued SHP2-null U2OS lines, knockout cells were infected with retrovirus harboring the wild-type or mutant SHP2 cDNAs and FACS sorted for GFP-positive cells. MDA-MB-468 parental cells stably expressing full-length GAB1-GFP, GAB2-GFP, SHP2, PTP, and N-SH2-C-SH2 cDNAs were also generated similarly. U2OS cell lines were grown in McCoy's 5A media with 10% fetal bovine serum (FBS). MDA-MB-468 cells were cultured in Leibovitz's L-15 media with 10% FBS without $CO_2$. All other cell lines (KYSE520, Jurkat, and NCI-H1975) were maintained in RPMI-1640 supplemented with 10% FBS at 5% $CO_2$. Cell lines were acquired from sources provided in the key resources table. All cell lines used in this study tested negative for mycoplasma contamination.

## Ligand stimulation

For ligand stimulation experiments analyzed by Western blotting, cells were seeded at 70% confluence in 10 cm petri dishes (Nunc, Thermo Fisher Scientific), serum starved for 24 hr, and treated with DMSO carrier or SHP099 (10 µM) for 2 hr before stimulation with ligands (EGF [10 nM], PDGF-BB [5 nM], or anti-CD3 [10 µg/ml]) for the indicated time periods. Cells were then quickly washed with ice-cold phosphate-buffered saline (PBS) twice, lysed in Ripa buffer (25 mM Tris–HCl [pH 7.6], 150 mM NaCl, 1% Nonidet P-40, 1% sodium deoxycholate, 0.1% sodium dodecyl sulfate, 2 mM EDTA) with protease and phosphatase inhibitors (cOmplete Mini and PhosSTOP, Roche) and analyzed by Western blotting. For all Western blot assays, the figures shown are representative results from at least two independent experiments.

## Sample preparation for phosphoproteomics

MDA-MB-468 cells were seeded in 15 cm cell culture dishes (Corning) in full growth media (Leibovitz's L-15 with 10% FBS and 1% penicillin/streptomycin). At 80% confluence, cells were washed twice with warm HBSS buffer and incubated in serum-free media for 24 hr. On the following day, cells were pre-treated with DMSO or SHP099 (10 µM) for 2 hr prior to ligand stimulation. As outlined in *Figure 1A*, DMSO and SHP099 groups were left untreated (0 min) or stimulated with EGF (10 nM) for 5, 10, or 30 min. The SHP099 washout group was stimulated with EGF (10 nM) for 10 min, washed three times with warm HBSS, and stimulated with EGF again for 5, 10, or 30 min. To terminate stimulation, cells were immediately washed with ice-cold PBS, harvested, centrifuged, flash frozen in liquid N2 and stored at −80°C until all 11 samples were prepared. Cell pellets were lysed in lysis buffer (2% SDS, 150 mM NaCl, 50 mM Tris [pH 8.5–8.8]) containing protease and phosphatase inhibitors (cOmplete Mini, PhosSTOP, Roche; 2 mM Sodium Orthovanadate, NEB), sonicated to shear chromatin, and centrifuged to remove cellular debris. A bicinchoninic acid (Pierce, Catalog # 23225) assay was performed according to manufacturer's instructions and lysates were normalized to a protein concentration of 1 mg/ml, reduced with DTT (5 mM), alkylated with iodoacetamide (14 mM), and quenched with further addition of DTT (5 mM). Total protein (1 mg) was precipitated with methanol-chloroform, reconstituted in 8 M urea at pH 8.5, sonicated, diluted to 4 M urea, and digested with Lys-C (FUJIFILM Wako, Catalog # 125–05061) overnight and Trypsin (Promega, Catalog # V511C) for 6 hr at an enzyme-to-protein ratio of 1:100. The percentage of missed cleavages was monitored using aliquots from a few representative samples (total 3 µg) that were desalted and analyzed by mass spectrometry. Digests were acidified with 2% formic acid and desalted using C18 Sep-Pak cartridges (WAT054960, Waters). Samples were dried by vacuum centrifugation, resuspended in 80% acetonitrile/0.15% trifluoroacetic acid (TFA), and normalized to 1 mg/ml for enrichment of phosphopeptides.

## Phosphopeptide enrichment

Phosphopeptides were enriched using IMAC with NTA Magnetic Agarose beads (Qiagen, Catalog # 36113) stripped of $Ni^{2+}$ and reloaded with Fe(III). Five hundred microliters of beads were incubated in 1 ml EDTA (40 mM) for 30 min at room temperature to remove bound $Ni^{2+}$, followed by three washes with 1 ml of HPLC-grade water. The beads were then charged with iron using $FeCl_3$ (100 mM) for 30 min at room temperature. Excess iron was removed by washing the beads three times with water followed by acidification with 80% acetonitrile/0.15% TFA. Peptides (1 mg per sample) were incubated with the IMAC beads for 30 min at room temperature. Phosphopeptide-bound beads were washed with 1 ml of 80% acetonitrile/0.15% TFA to remove non-specifically bound peptides. Phosphopeptides were eluted with 300 µl of 50% acetonitrile/0.7% $NH_4OH$, acidified with 4% formic acid, immediately vacuum centrifuged to dryness, and subjected to desalting using SOLA HRP 10 mg Sep-Pak cartridges (Thermo Fisher).

## TMT labeling

Prior to isobaric labeling, phosphopeptides were reconstituted in 200 mM EPPS (pH 8.3) and anhydrous acetonitrile to 30% (v/v). Each sample was labeled with 5 µl of a TMT 11-plex reagent (20 µg/µl; A37725, Thermo Scientific) for 90 min at room temperature. The reactions were quenched with 5 µl of 5% hydroxylamine for 15 min, combined, acidified with TFA to 1.0% (v/v), desalted using 10 mg SOLA Sep-Pak cartridges (Thermo Fisher), and lyophilized for 2 days to remove residual TFA.

## Phosphotyrosine immunoaffinity purification

The pY antibody (p-Tyr-1000, Cell Signaling Technology) was coupled with agarose beads one day prior to the immunoaffinity purification (IAP) as follows. A Protein A agarose bead slurry (Sigma–Roche) (60 µl) was washed four times with 1 ml of cold PBS. The p-Tyr-1000 antibody (Cell Signaling Technology) (30 µl) was gently mixed with 1.5 ml PBS and added onto the washed agarose beads and incubated on a rotator overnight at 4°C. The beads were washed four times in 1 ml of cold PBS to remove excess uncoupled antibody. Lyophilized peptides were reconstituted with 500 µl of IAP buffer (50 mM MOPS/NaOH pH 7.2, 10 mM $Na_2HPO_4$, 50 mM NaCl) and incubated with antibody conjugated beads for 2 hr at 4°C with gentle rotation. The non-specific binding phosphopeptide (phospho-serine and -threonine peptides) flow through was collected after centrifugation and stored at −80°C. The pY-bound peptides were washed once with 1 ml of IAP buffer and transferred into a 0.2 µM filter spin column followed by two 400 µl washes with cold HPLC-grade water. pY peptides were eluted twice with 75 µl of 100 mM formic acid. The pY eluate was desalted using a C18 Stage-Tip, dried by vacuum centrifugation, and reconstituted in 3% acetonitrile/0.5% formic acid for MS analysis.

## Mass spectrometry

Phosphoproteomic mass spectrometric data were acquired on an Orbitrap Fusion Lumos mass spectrometer (Thermo Fisher Scientific) equipped with a Proxeon EASY-nLC 1000 liquid chromatography system (Thermo Fisher Scientific). Phosphopeptides were separated on a 75 µm inner diameter microcapillary column packed with ~35 cm Sepax GP-C18 resin (1.8 µm, 150 A, Thermo Fisher Scientific). Phosphopeptides (~2 µg) were separated using a 2 hr gradient of acetonitrile in 0.125% formic acid.

The phosphoproteome MS analysis scan sequence began with the collection of an FTMS1 spectrum (120,000 resolution with mass range 400–1400 Th). The top 10 most intense ions were selected for MS/MS and fragmented via collision-induced dissociation (CID, CE = 35%) with a maximum injection time of 200 ms and an isolation window of 0.5 Da. FTMS3 precursors were fragmented by high-energy collision-induced dissociation (HCD, CE = 55%) and analyzed at 50,000 resolution (200 Th) with a maximum ion injection time of 300 ms and an isolation window of 1.2. The MultiNotch MS3-based TMT method was applied as described in *McAlister et al., 2014*.

## Data processing

Mass spectra were processed using a SEQUEST-based software pipeline. A modified version of ReAdW.exe was used to convert spectra (.raw) to mzXML. All spectra were searched against a database containing the human proteome downloaded from Uniprot (February 4, 2014) and common contaminating protein sequences. Database searches were performed using a peptide mass tolerance of 50 ppm and a fragment ion tolerance of 0.9 Da. TMT on lysine residues and peptide N termini (+229.163 Da) and carbamidomethylation of cysteine residues (+57.021 Da) were fixed modifications, while oxidation of methionine residues (+15.995 Da) and phosphorylation of serine, threonine, and tyrosine residues were set as a variable modification (+79.966 Da).

Peptide-spectrum matches were filtered by linear discriminant analysis to a false discovery rate (FDR) of 2% at the peptide level based on matches to reversed sequences. Linear discriminant analysis considered the following parameters: XCorr, ΔCn, missed cleavages, adjusted PPM, peptide length, fraction of ions matched, charge state, and precursor mass accuracy. Filtered peptides were collapsed further to a final protein-level FDR of 2%. Phosphopeptides were quantified from MS3 scans after filtering with a total TMT reporter signal-to-noise ratio > 200 and isolation specificity at 0.5. Localization of phosphorylation sites was determined using AScore (*Huttlin et al., 2010*), and sites with an AScore > 13 were selected for further analysis.

## Dependence classification of phosphorylation site time courses

Each phosphorylation time course was classified according to its response to (1) EGF stimulation and (2) SHP2 inhibition. This analysis was performed using the union of the two biological replicates, averaging measurements for sites that were found in the intersection of replicates. EGF classes were defined based on phosphorylation dynamics in control cells only. Significant increases and decreases were defined as changes of at least 1.5-fold relative to time 0 (unstimulated cells). The classes

comprised the following: (1) fast increase (increased phosphorylation within 5 min of EGF stimulation), (2) medium increase (increased phosphrylation within 10 min), (3) slow increase (increased phosphorylation within 30 min), (4) neutral (no change at any timepoint), and (5) decrease (decreased phosphorylation by any point in the time course).

## EGF classes

| Name | Earliest timepoint where change is detected |
|---|---|
| Fast increase | 5 min |
| Medium increase | 10 min |
| Slow increase | 30 min |
| Neutral | n/a |
| Decrease | Any |

Classes of SHP2 responses were defined based on comparison between control cells and cells treated with SHP099 for two hours prior to EGF stimulation. The classes comprised the following: (1) pre-stimulation negative (increased phosphorlyation in SHP099-treated cells compared to control cells, prior to EGF addition), (2) post-stimulation negative (increased phosphorylation in SHP099-treated cells at any point after EGF addition), (3) neutral (no difference between treatment and control at any timepoint), (4) pre-stimulation positive (decreased phosphorylation in SHP099-treated cells compared to control cells, prior to EGF addition), and (5) post-stimulation positive (decreased phosphorylation in SHP099-treated cells at any point after EGF stimulation).

## SHP2 classes

| Name | Change with SHP099 treatment | Timing of change |
|---|---|---|
| Pre-stimulation negative | Increase | Before EGF addition |
| Post-stimulation negative | Increase | Any timepoint after EGF addition, but not before |
| Neutral | None | n/a |
| Pre-stimulation positive | Decrease | Before EGF addition |
| Post-stimulation positive | Decrease | Any timepoint after EGF addition, but not before |

## Bioinformatic analysis

Volcano plots were generated for all sites that were detected in either of the biological replicates. For each site at each timepoint (5 min, 10 min, and 30 min), the mean of [SHP099/DMSO] and [SHP099/WO] fold-changes was plotted against –log10(p-value). Significantly altered phosphorylation sites were visualized based on their membership in the Reactome database (http://software.broadinstitute.org/gsea/msigdb/annotate.jsp, RRID:SCR_003485). Gene set analysis was performed using publicly available GSEA tools (RRID:SCR_003199) (*Mootha et al., 2003*; *Subramanian et al., 2005*), which were used to compute the overlap between detected proteins and the REACTOME pathway database. Hierarchical clustering was performed using the clustergram function of MATLAB (RRID:SCR_001622), with a Euclidean distance metric.

## Immunoprecipitation assays

MDA-MB-468 cells were seeded at 80% confluence in 10 cm dishes (Nunc, Thermo Fisher Scientific), serum starved for 24 hr, and treated with DMSO carrier or SHP099 (10 µM) for 2 hr before stimulation with EGF (10 nM) for 5 min, 10 min, or 30 min. Cells were then quickly washed with ice-cold PBS, lysed in 1 ml of RIPA buffer (25 mM Tris–HCl (pH 7.6), 150 mM NaCl, 1% Nonidet P-40,

1% sodium deoxycholate, 0.1% sodium dodecyl sulfate, 2 mM ethylenediaminetetraacetic acid [EDTA]) containing protease and phosphatase inhibitors (cOmplete Mini and PhosSTOP, Roche) on ice for 5 min and sonicated briefly to dissolve the pellet. After centrifugation at 16,000 g for 20 min, the lysates were pre-cleared with protein A/G agarose (20421, Pierce) for 1 hr at 4°C. Pre-cleared lysates were incubated with the specified antibodies overnight at 4°C. Immunoprecipitates were then pulled down by incubating lysates with protein A/G agarose for 1 hr at 4°C, washed four times with ice-cold RIPA buffer, eluted in protein loading buffer, and analyzed by Western blotting. For all immunoprecipitation – Western blotting assays, the figures shown are representative results from at least two independent experiments.

## In vitro phosphate release assay

Phosphate released from synthetic phosphopeptide substrates by SHP2 was measured using Malachite Green assay (MAK307, Sigma–Aldrich). The dephosphorylation reaction contained 50 nM wild-type SHP2(1–525), 6 µM bisphosphorylated IRS1 ligand for activating SHP2, and varying concentrations of phosphopeptide substrates (0–0.5 mM) in a final volume of 40 µl buffer (60 mM HEPES [pH 7.2], 75 mM KCl, 75 mM NaCl, 1 mM EDTA, 0.05% Tween 20, and 2 mM DTT). The reactions were incubated at room temperature for 5 min in a 384-well plate in triplicate and terminated by the addition of 10 µl malachite green solution (Sigma–Aldrich, Catalog # MAK307) to each well. After further incubation for 30 min at room temperature for color development, absorbance was measured at 620 nm on a plate reader. The amount of phosphate released was determined by comparison with a malachite green standard curve. The reaction rate, defined as micromoles of phosphate released per minute, reflects the dephosphorylation activity of SHP2 toward its phosphopeptide substrates. The sequences of the peptides used in the assay were:

> IRS1_pY1172–PEG–pY1222 – SLN{pY}IDLDLVK–dPEG8–LST{pY}ASINFQK
> OCLN_pY443 – TGLQE{pY}KSLQS
> PLCG2_pY818 – TRIQQ{pY}FPSNY
> GRB2_pY209 – MFPRN{pY}VTPVN
> ARHGAP35_pY1105 – EENI{pY}SVPHDS
> CSF-1R_pY708 – IHLEKK{pY}VRRDSGF
> TMEM134_pY57 – KQSRLR{pY}QNLEND

## Proximity ligation assay

MDA-MB-468 cells were grown to 70% confluence on eight-chambered glass slides (Nunc Lab-Tek II, Thermo Scientific), starved in serum-free media for 24 hr, and treated with DMSO carrier or SHP099 (10 µM) for 2 hr prior to stimulation with 1 nM EGF for 2 min. Cells were fixed with ice-cold 4% (v/v) paraformaldehyde for 15 min and permeabilized with 0.125% Tween-20 in PBS for 5 min prior to incubation in 1× blocking solution (1× PBS, 10% BSA, 10% donkey serum, 10% goat serum) for 1 hr at room temperature. Cells were then incubated with mouse anti-EGFR (1:1000, clone 13G8, Millipore) and rabbit anti-GAB1 (1:100, Catalog # SAB4501060, Sigma) overnight at 4°C in a humidified chamber. The proximity ligation assay was conducted using a DuoLink In Situ Mouse/Rabbit PLA kit (Sigma–Aldrich, Catalog # DUO92014) according to the manufacturer's instructions. All images were collected on a confocal laser scanning microscope (Zeiss LSM 880 with Airyscan) equipped with a 63× oil immersion objective lens (Plan Apo, NA = 1.4) and analyzed using CellProfiler software (RRID:SCR_007358). Three independent replicates (n = 3) were performed.

## Acknowledgements

We thank Jon Aster, Huaixiang Hao, Aram Ghalali, Abhinav Dubey, and members of the Blacklow laboratory for helpful discussions. Funding: This work was supported by NIH grants R35 CA220340 (SCB), U54-CA255088 (LAC and PKS), and from the Dana Farber-Novartis Translational Drug Development Program. AP was supported by the Novo Nordisk Foundation Copenhagen Bioscience PhD Programme grant (grant number NNF16CC0020906). Competing Interests: SCB receives research funding for this project from Novartis, is a member of the SAB of Erasca, Inc, is an advisor to MPM Capital, and is a consultant on unrelated projects for IFM, Scorpion Therapeutics, and Ayala Therapeutics. PKS is a member of the SAB or Board of Directors of Merrimack Pharmaceutical, Glencoe

Software, Applied Biomath, and RareCyte Inc and has equity in these companies. Sorger declares that none of these relationships are directly or indirectly related to the content of this manuscript. MGA, ML, and MM are former employees of Novartis. Data and materials availability: Quantitative proteomics data has been deposited in the mass spectrometry interactive virtual environment (MassIVE) database at: https://doi.org/doi:10.25345/C5HS6C

## Additional information

### Competing interests

Morvarid Mohseni, Matthew J LaMarche, Michael G Acker: Novartis employee while this work was performed. Stephen C Blacklow: SCB receives research funding for this project from Novartis, is a member of the SAB of Erasca, Inc, is an advisor to MPM Capital, and is a consultant on unrelated projects for IFM, Scorpion Therapeutics, and Ayala Therapeutics. The other authors declare that no competing interests exist.

### Funding

| Funder | Grant reference number | Author |
|---|---|---|
| Novartis | | Vidyasiri Vemulapalli<br>Khal-Hentz Gabriel<br>Jonathan LaRochelle<br>Kimberley Stegmaier<br>Stephen C Blacklow |
| National Cancer Institute | R35 CA220340 | Stephen C Blacklow |
| National Institutes of Health | U54-HL127365 | Lily A Chylek<br>Peter K Sorger |
| Novo Nordisk Foundation | NNF16CC0020906 | Anamarija Pfeiffer |
| National Cancer Institute | U54-CA225088 | Lily A Chylek |

The funders had no role in study design, data collection and interpretation, or the decision to submit the work for publication.

### Author contributions

Vidyasiri Vemulapalli, Data curation, Formal analysis, Validation, Investigation, Visualization, Methodology, Writing - review and editing; Lily A Chylek, Formal analysis, Methodology, Writing - review and editing, Visualization; Alison Erickson, Investigation, Methodology, Writing - review and editing; Anamarija Pfeiffer, Khal-Hentz Gabriel, Validation, Investigation; Jonathan LaRochelle, Conceptualization, Investigation, Methodology; Kartik Subramanian, Conceptualization; Ruili Cao, Resources, Supervision, Expressed and purified SHP2 protein for enzymatic assays; Kimberley Stegmaier, Supervision; Morvarid Mohseni, Matthew J LaMarche, Conceptualization, Supervision; Michael G Acker, Conceptualization, Supervision, Methodology, Writing - review and editing; Peter K Sorger, Conceptualization, Resources, Supervision, Methodology, Writing - review and editing; Steven P Gygi, Conceptualization, Resources, Supervision, Funding acquisition, Visualization, Methodology, Writing - original draft, Project administration, Writing - review and editing; Stephen C Blacklow, Conceptualization, Resources, Supervision, Funding acquisition, Visualization, Writing - original draft, Project administration, Writing - review and editing

### Author ORCIDs

Vidyasiri Vemulapalli [iD] https://orcid.org/0000-0002-3368-4766
Lily A Chylek [iD] https://orcid.org/0000-0003-1216-1250
Kartik Subramanian [iD] http://orcid.org/0000-0002-6900-8882
Peter K Sorger [iD] http://orcid.org/0000-0002-3364-1838
Stephen C Blacklow [iD] https://orcid.org/0000-0002-6904-1981

**Decision letter and Author response**

Decision letter https://doi.org/10.7554/eLife.64251.sa1

Author response https://doi.org/10.7554/eLife.64251.sa2

## Additional files

### Supplementary files

• Supplementary file 1. Tables showing all pY sites quantified in the first biological replicate of the phosphoproteomic screen.

• Supplementary file 2. Table showing all pY sites quantified in the second biological replicate of the phosphoproteomic screen.

• Supplementary file 3. Table displaying the 20 candidate substrate proteins tested.

• Transparent reporting form

### Data availability

Quantitative proteomics data have been deposited in the mass spectrometry interactive virtual environment (MassIVE) database with the accession code MSV000083702. All other data generated in this study are in the manuscript and supporting files.

The following dataset was generated:

| Author(s) | Year | Dataset title | Dataset URL | Database and Identifier |
|---|---|---|---|---|
| Vemulapalli V, Erickson AR, Gygi SP, Blacklow SC | 2021 | Time resolved phosphoproteomics reveals scaffolding and catalysis-responsive patterns of SHP2-dependent signaling | https://doi.org/10.25345/C5HS6C | MassIVE MSV0000 83702, 10.25345/C5HS6C |

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
