## [Decision Letter]

**Acceptance summary:**

The authors used phosphotyrosine proteomics to uncover novel aspects of signaling by PTPN11/SHP2, a phosphotyrosine phosphatase that contains two SH2 domains and is involved in Noonan's Syndrome and cancer. By creative use of knockouts, mutants, and a chemical inhibitor that locks SHP2 in a state where its SH2 domains are unavailable, the authors provide evidence for two classes of phosphoproteins, those that are SHP2 substrates and those that appear to be protected from other PTPs by the SHP2 SH2 domains. The results help explain how SHP2 has both positive and negative effects on signaling.

**Decision letter after peer review:**

Thank you for submitting your article "Time resolved phosphoproteomics reveals scaffolding and catalysis-responsive patterns of SHP2-dependent signaling" for consideration by *eLife*. Your article has been reviewed by 3 peer reviewers, and the evaluation has been overseen by a Reviewing Editor and Senior Editor. The following individual involved in review of your submission has agreed to reveal their identity: Bruce J Mayer (Reviewer #1).

The reviewers have discussed the reviews with one another and the Reviewing Editor has drafted this decision to help you prepare a revised submission.

Summary:

This manuscript addresses the importance of the Shp2 tyrosine phosphatase in shaping the response of cells to EGF stimulation. The profile of EGF-induced phosphotyrosine (pTyr)-containing peptide sites was measured with and without SHP099, a Shp2 allosteric inhibitor. As expected, the abundance of a number of phosphopeptides was increased by inhibitor treatment, consistent with being Shp2 substrates. In contrast, the phosphorylation of a number of peptides, such as those on Gab1/2, was decreased upon Shp2 inhibition. The authors go on to present evidence that this is likely due to phosphosite protection by the Shp2 SH2 domains.

This is a technically outstanding piece of work that makes extensive use of quantitative mass spectrometry. The concepts of SH2 protection and of Shp2 scaffold function are not novel, and conclusions regarding potential new Shp2 substrates are over-interpreted, but the results will be of general interest to the field due to the importance of Shp2 in stimulating RTK-mediated MAPK and PI3K pathways. The implication of Shp2 in important and cancer-relevant signaling pathways increases potential impact. That said, the value of this study is more in developingig new testable hypotheses rather than in providing new insights into the actual signaling mechanisms. The review consensus is that paper would be acceptable with only minor experimental work and rewriting, but would be much more interesting with deeper investigation of the proposed new substrates or the scaffolding functions.

Essential revisions:

1. Figure 4F: While we commend the authors for showing that Shp2 can dephosphorylate occludin, ARHGAP35 and PLCg2 peptides, additional work would be needed to conclusively demonstrate that these are indeed natural substrates. Does Shp2 interact with any of these proteins in cells and in cell-free systems? Does the enzymatic dead mutation affect the interaction? In the phosphatase enzymatic assay, it is nice to include CSF-1R substrate as a negative control. How about a type III peptide? Would the enzymatic dead mutant of SHP2 have any impact on phosphate release? What were the 20 candidate proteins initially screened? Were any of these 20 candidates established Shp2 substrates? Overall, the conclusions relating to potential Shp2 substrates are over interpreted. Adding more experimental information would be desirable, but, failing that, the authors should temper their conclusions.

2. As interesting as the SH2 phosphosite protection may be, the actual role of this phenomenon in Shp2 signaling is not really clarified. In other words, is protection merely a consequence of Shp2 recruitment, or does it have functional consequences beyond bringing the catalytic activity of Shp2 to membrane signaling complexes? The authors discuss potential scaffolding functions, but it is not clear what they mean-do they propose other signaling proteins are bound to other domains/sites of Shp2 that are critical for downstream signaling? The Shp2 mutants and knockout cell lines used here would be ideal to test a few simple ideas. For example, to what extent can the EK/CE (open/dead) mutant rescue EGF-dependent downstream signaling compared to the EK (open/active) mutant? Similarly, does the Shp2 SH2-only construct have any biological activity when compared to EK/CE? The impact of the manuscript would be much greater if some insight into the role of SH2 protection in Shp2 signaling were gained.

3. Regarding potential scaffold function: phosphorylation of Shp2 itself is increased when the Shp2 inhibitor is removed, falling into the same category as GAB Y659. Given the potential role of Shp2 phosphorylation in recruiting Grb2, the authors should discuss the possible scaffolding role of the Shp2 pY sites and their interpretation of why phosphorylation of those sites increases when Shp2 inhibitor is removed.

4. The EGF stimulation time course included only 5- 10- and 30-min time points. This misses a whole class of interesting/important phosphorylation events that occur very rapidly after EGF stimulation. While MAPK may take ~5 minutes to be fully activated, many pTyr sites are maximally phosphorylated within one or two minutes of EGF treatment, and in particular Shp2 SH2 binding sites have been reported to peak within a minute or two and then return close to baseline by 5 minutes (e.g. Jadwin et al., *eLife* 2016;5:e11835). It is not realistic to expect the authors to repeat the MS experiments with earlier timepoints, but it would be relatively trivial to provide overall anti-pTyr blots for the timepoints used for MS (to gauge the extent of overall pTyr changes with and without EGF, Shp2 inhibitor in these cells), and more importantly to provide more detailed timecourses for selected "SH2 protected" substrates such as GAB1 or EGFR with and without Shp2 inhibition. In addition, for the MAPK analysis, the time course should be extended to 2-4 hrs to include the decay period. The experiments should be quantitated (pERK/ERK).

5. The interpretation that MPZL and the GAB sites were being reduced in their level of phosphorylation as a result of SH2 domain protection is misleading and in its present form still open to the possibility raised by the authors that this is a Src-mediated phenomenon. There was no change in overall Src activity assayed using the pY416 antibody in total cell lysates but relevant changes in Src activity may be restricted to the precise protein complex in which it is modulated. One would need to determine the status of Src activation not in a whole cell lysate but rather the status of Src pY416 in the GAB complex. This is a well-established behavior of the SFKs, in general. Similar experiments should be performed with PZR in complex with Src as similar mechanisms are likely operative where Src can phosphorylate PZR, which has been shown, thereby creating binding sites for SH2 domains of Shp2. Deletion of the SH2 domains of Shp2, which binds Src, would fail to recruit Src to PZR resulting in reduced phosphorylation. These possibilities are all arguable and not in of themselves as presented "air-tight".

---

## [Author Response]

Essential revisions:1. Figure 4F: While we commend the authors for showing that Shp2 can dephosphorylate occludin, ARHGAP35 and PLCg2 peptides, additional work would be needed to conclusively demonstrate that these are indeed natural substrates. Does Shp2 interact with any of these proteins in cells and in cell-free systems? Does the enzymatic dead mutation affect the interaction? In the phosphatase enzymatic assay, it is nice to include CSF-1R substrate as a negative control. How about a type III peptide? Would the enzymatic dead mutant of SHP2 have any impact on phosphate release? What were the 20 candidate proteins initially screened? Were any of these 20 candidates established Shp2 substrates? Overall, the conclusions relating to potential Shp2 substrates are over interpreted. Adding more experimental information would be desirable, but, failing that, the authors should temper their conclusions.

We focused attention on potential substrates identified from the mass spectrometry data. We identified pY sites on 20 proteins (now included in the manuscript as Supplementary file 3) that met the following criteria:

1. The identified site was not previously reported as a SHP2 substrate

2. The pY abundance of the site was increased more than two-fold in the presence of SHP099 when compared to DMSO control (*i.e*. [SHP099/DMSO] fold-change > 2)

3. The pY abundance of the site is increased in response to EGF stimulation

4. Availability of commercial antibodies to the identified protein

We also used Y209 of GRB2 as a positive control for these studies because it is a well-established SHP2 substrate that met the other criteria (2-4) above. For 13 proteins, antibody sensitivity was insufficient to determine whether they are SHP2 substrates. For four proteins, Western blotting of the immunoprecipitates did not show increased pY abundance upon SHP099 treatment. The three proteins that did show increased pY abundance in response to SHP099 were occludin, ARHGAP35, and PLCγ2. The text has been modified on page 9 to include this information.

The conclusion that occludin, ARHGAP35, and PLCγ2 are substrates for SHP2 include both immunoprecipitation data showing that they have an increased amount of pY in the presence of SHP099 and in vitro confirmation that a pY-containing peptide spanning the site identified by MS is active as a substrate for the purified enzyme (Figure 3F). In contrast and as expected, a purified catalytically inactive mutant form of the enzyme (C459A), does not dephosphorylate these substrate phosphopeptides (wt and mutant data are plotted in Author response image 1).

**Author response image 1. sa2fig1:** Dephosphorylation activity of wild-type or C459A SHP2, activated with 6 μM bisphosphorylated IRS-1 peptide [SLNY(p)IDLDLVKdPEG8-LSTY(p)ASINFQK], towards synthetic phosphopeptide substrates (OCLN_pY443, PLCG2_pY818, GRB2_pY209, ARHGAP35_1105, TMEM134_pY57, CSF-1R_pY708). The C459A mutant is inactive in the dephosphorylation assay, as expected.

We did test whether SHP2 can be co-immunoprecipitated with occludin, ARHGAP35, or PLCγ2 in MDA-MB-468 cells, but did not see changes in bulk immunoprecipitation of any of these proteins with SHP2 after EGF stimulation. We note, however, that bulk recovery of a protein by co-IP is not a prerequisite for a bona-fide protein substrate.

Lastly, we are not certain what the reviewer meant by the term “Type III” peptide, but we interpreted the meaning as equivalent to what we defined in the manuscript as “Class III” peptides. The ability of SHP2 to dephosphorylate pY57 of TMEM134, a “Class III” peptide, is greatly reduced by comparison to the putative substrate peptides. These data are now included in the results reported in Figure 3F, and the text on pages 9-10 has been modified to include these new data. Lastly, as recommended, we have softened the concluding sentence of the paragraph to state that our results “suggest,” rather than “support the conclusion,” that ARHGAP35, PLCγ2, and occludin are newly identified substrates for SHP2.

2. As interesting as the SH2 phosphosite protection may be, the actual role of this phenomenon in Shp2 signaling is not really clarified. In other words, is protection merely a consequence of Shp2 recruitment, or does it have functional consequences beyond bringing the catalytic activity of Shp2 to membrane signaling complexes? The authors discuss potential scaffolding functions, but it is not clear what they mean-do they propose other signaling proteins are bound to other domains/sites of Shp2 that are critical for downstream signaling? The Shp2 mutants and knockout cell lines used here would be ideal to test a few simple ideas. For example, to what extent can the EK/CE (open/dead) mutant rescue EGF-dependent downstream signaling compared to the EK (open/active) mutant? Similarly, does the Shp2 SH2-only construct have any biological activity when compared to EK/CE? The impact of the manuscript would be much greater if some insight into the role of SH2 protection in Shp2 signaling were gained.

We agree with the reviewer that elucidating the functional role of phosphosite protection by the SH2 domains of SHP2 is an interesting future direction. While it is beyond the scope of the current work to investigate the functional implications of this phenomenon in detail, there is some evidence that the scaffolding/protective activity of the SH2 domains has a functional effect from the results of experiments performed with the SH2 domain-only SHP2 fragment (Figure 5D). The data show accumulation of protected pY marks on both GAB and GAB2 even in the absence of EGF stimulation, consistent with a dominant effect of the tandem SH2 domain fragment that is independent of the catalytic domain of the enzyme. This observation is now noted on Page 11 of the manuscript.

3. Regarding potential scaffold function: phosphorylation of Shp2 itself is increased when the Shp2 inhibitor is removed, falling into the same category as GAB Y659. Given the potential role of Shp2 phosphorylation in recruiting Grb2, the authors should discuss the possible scaffolding role of the Shp2 pY sites and their interpretation of why phosphorylation of those sites increases when Shp2 inhibitor is removed.

The presence of the allosteric inhibitor locks SHP2 in the closed conformation. Thus, the inhibitor interferes with membrane recruitment of SHP2 because its SH2 domains are not available to bind to pY residues deposited at membrane proximal sites by activated kinases. It is thus reasonable to postulate that removal of inhibitor permits membrane recruitment to sites of EGFR activation, phosphorylation of the SHP2 C-terminal tail, and subsequent GRB2 recruitment. We have added text discussing this issue to the manuscript on pages 12-13.

4. The EGF stimulation time course included only 5- 10- and 30-min time points. This misses a whole class of interesting/important phosphorylation events that occur very rapidly after EGF stimulation. While MAPK may take ~5 minutes to be fully activated, many pTyr sites are maximally phosphorylated within one or two minutes of EGF treatment, and in particular Shp2 SH2 binding sites have been reported to peak within a minute or two and then return close to baseline by 5 minutes (e.g. Jadwin et al., eLife 2016;5:e11835). It is not realistic to expect the authors to repeat the MS experiments with earlier timepoints, but it would be relatively trivial to provide overall anti-pTyr blots for the timepoints used for MS (to gauge the extent of overall pTyr changes with and without EGF, Shp2 inhibitor in these cells), and more importantly to provide more detailed timecourses for selected "SH2 protected" substrates such as GAB1 or EGFR with and without Shp2 inhibition. In addition, for the MAPK analysis, the time course should be extended to 2-4 hrs to include the decay period. The experiments should be quantitated (pERK/ERK).

As noted by the reviewer (see both Jadwin et al., *eLife* 2016;5:e11835 and Reddy et al., PNAS 2016;113:3114), we do appreciate that the kinetics of EGF stimulated tyrosine phosphorylation rapidly builds to a peak within a minute or two. Because the arrival of pY binding proteins at the membrane, such as GRB2 (and presumably other pY binders like SHP2; Jadwin et al., 2016), is delayed relative to the deposition of the pY mark, reaching a peak after ~5-10 min, we therefore chose 5, 10, and 30 minute timepoints for the mass spec measurements to maximize readout of the role of SHP2 downstream of EGF stimulation. This point is now noted on page 5 of the manuscript.

As requested by the reviewers, we now include quantification of the data in Figure 1A (right panel). We also provide more extended timecourse data that includes earlier timepoints (2 min EGF) and additional timepoints through 4 hours in the absence and presence of SHP099 for ERK1/2 (T202/Y204 of ERK1, T185/Y187 of ERK2), GAB1 (pY627 and pY659), GAB2 (pY614 and pY643), and phospho-tyrosine (P-Tyr-1000, CST). The ERK1/2 and pTyr-1000 data are reported in Figure 1—figure supplement 1A, and the GAB1 and GAB2 are presented in Figure 4C (replacing the prior figure panel, which did not show timecourse data), along with changes in the text on page 10.

5. The interpretation that MPZL and the GAB sites were being reduced in their level of phosphorylation as a result of SH2 domain protection is misleading and in its present form still open to the possibility raised by the authors that this is a Src-mediated phenomenon. There was no change in overall Src activity assayed using the pY416 antibody in total cell lysates but relevant changes in Src activity may be restricted to the precise protein complex in which it is modulated. One would need to determine the status of Src activation not in a whole cell lysate but rather the status of Src pY416 in the GAB complex. This is a well-established behavior of the SFKs, in general. Similar experiments should be performed with PZR in complex with Src as similar mechanisms are likely operative where Src can phosphorylate PZR, which has been shown, thereby creating binding sites for SH2 domains of Shp2. Deletion of the SH2 domains of Shp2, which binds Src, would fail to recruit Src to PZR resulting in reduced phosphorylation. These possibilities are all arguable and not in of themselves as presented "air-tight".

Point appreciated. It is certainly possible that the SFK activity in the GAB complex is not reflected by assessment of bulk SFK activity in the whole cell lysate. We have now noted this caveat in the manuscript on page 13, and tempered our conclusions as recommended by the reviewers. Further studies investigating the effect of SHP2 inhibition on SFK would lie beyond the scope of this manuscript.